# The Role of Global Labels in Few-Shot Classification and How to Infer Them

**Ruohan Wang**
University College London
Institute for Infocomm Research, A*STAR
wang_ruohan@i2r.a-star.edu.sg

**Massimiliano Pontil**
Isitituto Italiano di Tecnologia
University College London
massimiliano.pontil@iit.it

**Carlo Ciliberto**
University College London
c.ciliberto@ucl.ac.uk

## Abstract

Few-shot learning is a central problem in meta-learning, where learners must quickly adapt to new tasks given limited training data. Recently, feature pre-training has become a ubiquitous component in state-of-the-art meta-learning methods and is shown to provide significant performance improvement. However, there is limited theoretical understanding of the connection between pre-training and meta-learning. Further, pre-training requires *global labels* shared across tasks, which may be unavailable in practice. In this paper, we show why exploiting pre-training is theoretically advantageous for meta-learning, and in particular the critical role of global labels. This motivates us to propose **Me**ta **La**bel Learning (MeLa), a novel meta-learning framework that automatically infers global labels to obtains robust few-shot models. Empirically, we demonstrate that MeLa is competitive with existing methods and provide extensive ablation experiments to highlight its key properties.

## 1 Introduction

A central problem in meta-learning is *few-shot learning* (FSL), where new tasks must be learned quickly given limited amount of training data. FSL has drawn increasing attention recently due to the high cost of collecting and annotating large datasets. To tackle the challenge of model generalization in FSL, meta-learning leverages past experiences of solving related tasks by directly learning transferable knowledge over a collection of FSL tasks. A diverse range of meta-learning methods tailored for FSL have been proposed, including optimization-based [e.g. 1, 5, 27], metric learning [e.g. 20, 21, 25], and model-based methods [e.g. 7, 14, 19]. The diversity of the existing strategies raises a natural question: do these methods share any common lessons for improving model generalization and for designing future methods?

Several papers addressed the above question. Chen *et al.* [2] identified that data augmentation and deeper network architecture significantly improve generalization performance across several meta-learning methods. On the other hand, Tian *et al.* [23] investigated a simple yet competitive approach: a linear model on top of input embeddings learned via feature pre-training. This approach ignores task structures from meta-learning and merges all tasks into a "flat" dataset of labeled samples. The desired input embedding is then learned by classifying all classes of the flat dataset.

Extensive empirical evidences supports the efficacy of feature pre-training in FSL. Pre-training alone already outperforms various meta-learning algorithms [23]. Recently, it is used as a ubiquitous

35th Conference on Neural Information Processing Systems (NeurIPS 2021).

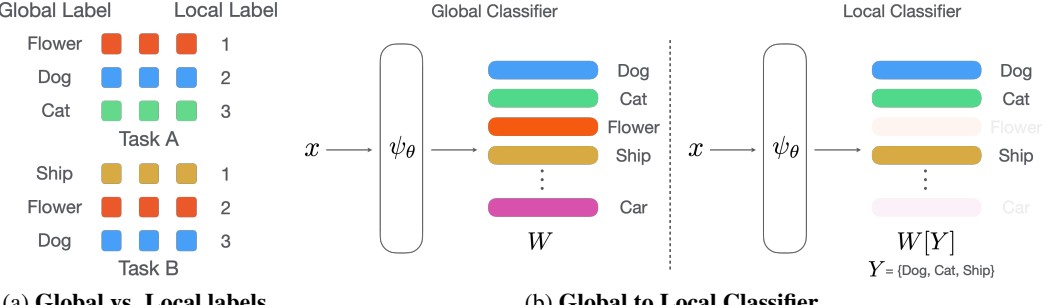

(a) **Global vs. Local labels**                    (b) **Global to Local Classifier**

Figure 1: **(a)** Colored squares represent samples. Tasks A and B can be "merged" meaningfully using global labels, but not local ones. **(b)** A global classifier can be used as local classifiers given the indices $Y$ of the intended classes to predict.

pre-processing step in state-of-the-art meta-learning methods [e.g. 18, 26, 27]. In particular, [27] reported that pre-training also significantly boosted earlier methods (see Fig. 2).

Despite the significant empirical improvement, there is limited theoretical understanding of why feature pre-training provides significant performance gain for meta-learning. On the other hand, pre-training requires task merging to construct the flat dataset, which implicitly assumes access to some notion of *global labels* consistently adopted across tasks. However, global labels may not exist or are inaccessible, such as when each task is annotated independently with only *local labels*. This renders direct task merging and consequently pre-training impossible (see Fig. 1a). Independent annotation captures realistic scenarios when tasks are collected *organically* (e.g from different users), rather than generated *synthetically* from benchmark datasets (e.g. *mini*IMAGENET). Possible scenarios include non-descript task labels (e.g. tasks with numerical labels) or even concept overlaps among labels across tasks (e.g. sea animals vs mammals).

In this paper, we address the issues raised above with the following contributions:

- In Sec. 3, we show that feature pre-training directly relates to meta-learning as a loss upper bound. In particular, pre-training induces a conditional meta-learning formulation [3, 26], which provides a principled explanation for the empirical performance gains.

- In Sec. 4, we propose **Me**ta **La**bel Learning (MeLa), a novel framework that automatically infers some notion of *latent* global labels consistent with local task constraints. The inferred labels enable us to leverage pre-training to improve meta-learners' generalization performance, and bridges the gap between experiment settings with or without acesss to global labels.

- In Sec. 5, we demonstrate empirically the competitive performance of MeLa over a suite of benchmark datasets. We also present ablation studies to highlight its key properties.

In Sec. 2, we first review the key notions of meta-learning and FSL. The supplementary material contains proofs for the theoretical analysis, additional empirical results, and experiment setup.

## 2   Background

We formalize FSL in the context of meta-learning, followed by reviewing feature pre-training initially investigated in [23].

**Meta-learning.** FSL [4] considers a meta-training set of tasks $\mathcal{T} = \{(S_t, Q_t)\}_{t=1}^{T}$, with *support set* $S_t = \{(x_j, y_j)\}_{j=1}^{n_s}$ and *query set* $Q_t = \{(x_j, y_j)\}_{j=1}^{n_q}$ sampled from the same distribution. Typically, $S_t$ and $Q_t$ each contains a small number of samples $n_s$ and $n_q$ respectively (fixed for all tasks for simplicity). We denote by $\mathcal{D}$ the space of datasets of the form $S_t$ or $Q_t$.

FSL aims to find the best inner learning algorithm (referred to as *base learner* in the following) $\mathrm{Alg}(\theta, \cdot) : \mathcal{D} \to \mathcal{F}$ that maps supports sets $S$ to predictors $f = \mathrm{Alg}(\theta, S)$, such that $x \mapsto f(x)$ generalizes well on the corresponding query sets. The base learner is meta-parametrized by $\theta \in \Theta$.

Formally, the meta-learning problem for FSL is

$$\min_{\theta \in \Theta} \ \mathbb{E}_{(S,Q) \in \mathcal{T}} \ \mathcal{L}\big(\mathrm{Alg}(\theta, S), \ Q\big), \tag{1}$$

where $\mathbb{E}_{(S,Q) \in \mathcal{T}} \triangleq \frac{1}{|\mathcal{T}|} \sum_{(S,Q) \in \mathcal{T}}$ denotes the empirical distribution over the meta-training set. The task loss $\mathcal{L} : \mathcal{F} \times \mathcal{D} \to \mathbb{R}$ is the empirical risk of the learner over query sets, according to an inner loss $\ell : \mathcal{Y} \times \mathcal{Y} \to \mathbb{R}$, where $\mathcal{Y}$ is the space of labels

$$\mathcal{L}(f, D) \ = \ \mathbb{E}_{(x,y) \in D} \ [\ell(f(x), y)]. \tag{2}$$

Designing effective base learners is a key focus in meta-learning literature and various strategies have been explored. We focus on a broad class of methods that we call meta-representation learning [1, 6, 10, 15], which is remarkably effective in practice and closely related to feature pre-training. Meta-representation learning infers a suitable embedding for base learners by solving

$$\min_{\theta \in \Theta} \ \mathbb{E}_{(S,Q) \in \mathcal{T}} \ [\mathcal{L}(w(\psi_\theta(S)), \psi_\theta(Q)] \tag{3}$$

where $\psi_\theta : \mathcal{X} \to \mathbb{R}^m$ is a feature embedding model and $\psi_\theta(D) \triangleq \{(\psi_\theta(x), y)|(x, y) \in D\}$ the embedded dataset. In [1], the ridge regression estimator is chosen as the base learner $\mathrm{Alg}(\theta, D) = w(\psi_\theta(D))$

$$w(\psi_\theta(D)) \ = \ \underset{W}{\mathrm{argmin}} \ \mathbb{E}_{(x,y) \in \psi_\theta(D)} \ \|Wx - \mathrm{OneHot}(y)\|^2 + \lambda_1 \|W\|^2, \tag{4}$$

where $\lambda_1$ is a constant and $\mathrm{OneHot}(y)$ denotes the one-hot encoding of the label $y$. Among different base learner designs, ridge regression is often favored for its differentiable closed-form solution and the associated computational efficiency in optimizing (3).

**Feature Pre-training.** Feature pre-training has been widely used in meta-learning. It was investigated in-depth in [23]. Given the meta-training set $\mathcal{T}$, a "flat" dataset $D_{\mathrm{global}}$ is constructed by merging all tasks in $\mathcal{T}$:

$$D_{\mathrm{global}} = D(\mathcal{T}) = \{(x_i, y_i)\}_{i=1}^N = \bigcup_{(S,Q) \in \mathcal{T}} (S \cup Q). \tag{5}$$

We then learn a embedding function $\psi_\theta$ on $D_{\mathrm{global}}$ using the standard cross-entropy loss $\ell_{\mathrm{ce}}$ for multi-class classification:

$$\underset{\theta, W}{\mathrm{argmin}} \, \mathbb{E}_{(x,y) \in D_{\mathrm{global}}} \ [\ell_{\mathrm{ce}}(W\psi_\theta(x), y)]. \tag{6}$$

After obtaining $\psi_\theta$, a novel task may be solved in the embedded space by a base learner. In [23], the authors recommended the regularized logistic regression estimator

$$w_{\mathrm{ce}}(D) = \underset{W}{\mathrm{argmin}} \ \mathcal{L}_{\mathrm{ce}}(W, D) + \lambda_2 \|W\|^2, \tag{7}$$

for its empirical performance. Here $\lambda_2$ is a regularization constant.

[23] demonstrated that feature pre-training as a standalone method already outperforms many sophisticated meta-learning strategies. It has also become a ubiquitous component in most state-of-the-art methods [27, 28, 29]. Furthermore, [27] demonstrated empirically that pre-training provides similar performance gain to earlier methods, making them (mostly) competitive with the state of the art. In Sec. 3, we show that feature pre-training directly relates to meta-learning as a loss upper bound, and explains why pre-training contributes to improved performance.

## 3 Feature Pre-training as Meta-learning

In this section, we show how feature pre-training relates to meta-learning as a loss upper bound. More precisely, we show that the pre-training induces a special base learner with the corresponding meta-learning loss upper bounded by the cross-entropy loss (Eq. (6)). Consequently, minimizing the cross-entropy loss also indirectly solves the induced meta-learning problem. Further, we observe that the special base learner is a conditional meta-learning formulation, which provides a principled explanation for the improved performance.

Let $\mathcal{T} = \{(S_t, Q_t)\}_{t=1}^T$ be a meta-training set. We denote the collection of query sets as $\mathcal{Q} = \{Q_t\}_{t=1}^T = \{(X_t, Y_t)\}_{t=1}^T$ where we write $Q_t = (X_t, Y_t)$ as a tuple of input samples $X_t = \{x_{jt}\}_{j=1}^{n_q}$ and their corresponding labels $Y_t = \{y_{jt}\}_{j=1}^{n_q}$. For simplicity, we assume that the query sets are disjoint, namely $Q_t \cap Q_{t'} = \emptyset$ for any $t \neq t'$. We merge all query sets into a flat dataset $D(\mathcal{Q}) = \{(x_i, y_i)\}_{i=1}^N = \cup_{t=1}^T Q_t$, with $N = n_q T$.

**Proposition 1.** *With the notation and assumptions introduced above, let $C$ be the total number of classes in $D(\mathcal{Q})$, and $W \in \mathbb{R}^{C \times m}$ the global classifier. Denote by $W[Y]$ the sub-matrix with rows indexed by the sorted unique values[1] from $Y$. Then, for any embedding $\psi_\theta : \mathcal{X} \to \mathbb{R}^m$*

$$\mathbb{E}_{(X,Y) \in \mathcal{Q}} \Big[ \mathcal{L}_{\text{ce}}\big(W[Y], (\psi_\theta(X), Y)\big) \Big] \leq \mathbb{E}_{(x,y) \in D(\mathcal{Q})} \left[ \ell_{\text{ce}}(W\psi_\theta(x), y) \right]. \tag{8}$$

We outline the proof strategy here and defer the details to the supplementary material. We first observe that $D(\mathcal{Q})$ has the same collection of samples as $\mathcal{Q}$. Secondly, each matrix $W[Y]$ is in fact a task classifier for the query set $(X, Y)$. Crucially, the likelihood of query samples $p(Y|X)$, estimated by task classifiers $W[Y]$, are no smaller than their likelihood estimated by the global classifier $W$, which forms the inequality in (8). Combining the two observations yields Prop. 1.

The key implication of Prop. 1 is that, *if global labels are available*, we can use a special base learner

$$w_{\text{global}}(D) = w_{\text{global}}(X, Y) = W[Y], \tag{9}$$

for any dataset of the form $D = (X, Y)$. This base learner only depends on the intended classes (i.e. unique values in $Y$) to produce task classifiers and is independent of specific inputs from $D$ (illustrated in Fig. 1b). We note that for any task $(S, Q) \in \mathcal{T}$, $w_{\text{global}}(S) = w_{\text{global}}(Q)$, since the support and query sets share the same class labels. These observations directly imply the following,

**Lemma 2.** *For meta-training set $\mathcal{T}$ where tasks are annotated with global labels, we have*

$$\mathbb{E}_{(S,Q) \in \mathcal{T}} \Big[ \mathcal{L}_{\text{ce}}\big(w_{\text{global}}(S), \psi_\theta(Q)\big) \Big] = \mathbb{E}_{(X,Y) \in \mathcal{Q}} \Big[ \mathcal{L}_{\text{ce}}\big(W[Y], (\psi_\theta(X), Y)\big) \Big]. \tag{10}$$

The left-hand side of (10) is thus the meta-learning loss associated with base learner $w_{\text{global}}(\cdot)$. Hence, the combination of Prop. 1 and Lemma 2 implies that the empirical risk associated with this meta-learning loss is upper bounded by the standard cross-entropy loss, namely the right-hand side of (8). The connection shows that feature pre-training also solves a meta-learning problem when we design the base learner to be $w_{\text{global}}(\cdot)$.

**Remark 1.** *The bound in Prop. 1 is tight when the standard cross-entropy loss is 0.*

Remark 1 shows that all task classifiers $W[Y]$ and the embedding function $\psi_\theta$ are optimal, when standard cross-entropy is 0. In practice, Remark 1 is achievable since over-parametrized neural networks could obtain zero empirical loss.

We highlight two important properties for the base learner $w_{\text{global}}(\cdot)$. Firstly, $w_{\text{global}}$ is not the base learner used during meta-testing since it cannot generalize to novel classes. In [23], $w_{\text{global}}$ was simply replaced with Eq. (7) during meta-testing, while other works choose to fine-tune the pre-trained embedding model $\psi_\theta$ with a new base learner intended for meta-testing using episodic training. The advantage of the fine-tuning strategy is clear since it optimizes $\psi_\theta$ for the actual base learner used during test time. This observation is well supported by existing empirical results, with many state-of-the-art methods [e.g. 28, 29] adopting the fine-tuning strategy and surpassing standalone pre-training.

Secondly, we observe that $w_{\text{global}}(\cdot)$ describes a conditional meta-learning problem: the global labels $Y$ provide additional side information to facilitating model learning. Specifically, global labels directly reveal how task samples relate to one another and $w_{\text{global}}$ could simply map global labels to task classifiers via $W[Y]$. In contrast, unconditional base learners (e.g. (4) and (7)) have to learn classifiers based on support sets, without access to task relations provided by global labels.

Global labels offer significant advantages to learning the embedding model $\psi_\theta$. Denevi *et al.* [3] proved that conditional meta-learning is advantageous over unconditional formulation by incurring a smaller excess risk, especially when the meta-distribution of tasks is organized into distant clusters

---

[1]E.g. $Y = \{6, 6, 4, 9\}$ maps to $\{4, 6, 9\}$-th rows of $W$. Also see Fig. 1b for an illustration.

(see [3] for further discussion). In practice, global labels cluster task samples for free and improve regularization by enforcing each cluster (denoted by global label $y$) to share vector $W[y]$ for all task classifiers. The above analysis explains why pre-training yields significantly more robust $\psi_\theta$ than many (unconditional) meta-learning methods.

Algorithm 1: **MeLa**

---

**Input:** meta-training set $\mathcal{T} = \{S_i, Q_i\}_{i=1}^T$
  $\psi_\theta^0 = \operatorname{argmin}_{\psi_\theta} \mathbb{E}_{(S,Q) \in \mathcal{T}} \left[ \mathcal{L}(w(\psi_\theta(S), \psi_\theta(Q))) \right]$
  Global clusters $G = \text{LearnLabeler}(\psi_\theta^0, \mathcal{T})$
  $\psi_\theta^* = \text{MetaLearn}(G, \mathcal{T})$
  **Return** $\psi_\theta^*$

---

Figure 2: Effect of Pre-training on *mini*IMAGENET

In Fig. 2, we plot several existing results from [2, 27] to highlight the contribution of the pre-training. For *mini*IMAGENET, pre-training accounted for on average 9% improvement in ProtoNet and MatchNet. In addition, pre-training alone yields a performance mostly competitive with both ProtoNet and MatchNet, suggesting that the pre-training contributes far more towards generalization performance compared to the specific meta-learning strategies deployed. Similar trends are observed for other methods and datasets, both in the previous works and our experiments. The extensive empirical results are consistent with the theoretical advantages of conditional meta-learning.

## 4 Meta Label Learning

In Sec. 3, we analyzed the theoretical advantages of feature pre-training in connection to meta-learning, as well as the critical role of global labels in learning robust embedding model. However, we argue that leveraging global labels are problematic for meta-learning. Firstly, **global labels may be unavailable or inaccessible in practical applications**, when meta-training tasks are collected and annotated independently as discussed in Sec. 1, rendering pre-training inapplicable. Secondly, **global labels oversimplify meta-learning**: they directly reveal how input samples relate to one another across tasks, while the goal of meta-learning is precisely learning such cross-task relations and extracting transferable knowledge. As highlighted in Sec. 3, pre-training contributes more towards test performance than the meta-learning strategies employed, making it difficult to assess the relative merits of different strategies.

To address these issues, we present a novel meta-learning framework that does not require access to global labels. In particular, our framework automatically infers some notion of latent global labels across tasks, therefore bridging the experiment settings with and without global labels.

Alg. 1 outlines our proposed strategy. We first meta-learn an embedding function $\psi_\theta^0$ as a tool to measure sample similarity. Secondly, we introduce a labeling algorithm for clustering task samples while enforcing local task constraints in the training data. The resulting clusters are used as inferred global labels. Lastly, we may apply any existing meta-learning strategy capable of leveraging global labels to obtain the final model[2].

We focus on the labeling algorithm since other components of MeLa are standard procedures. The labeling algorithm takes a meta-training set as input and outputs a set of clusters to represent global labels. The algorithm consists of a clustering step for updating centroids and a pruning step for merging small clusters. The algorithm is presented in Alg. 2.

**Clustering Step.** The procedure exploits local labels from each task to guide sample assignments. For each task, local labels are used to enforce two constraints: samples with the same local label should be assigned the same global label, while samples from different local classes should be assigned different global ones. Formally, given a set of cluster centroids $G = \{g_j\}_{j=1}^J$, we assign all

---

[2]We note that [9] is a loosely similar strategy: they use a fully unsupervised feature representation for clustering to infer auxiliary labels, which are then used to refine the representation.

---

**Algorithm 2** LearnLabeler

---

**Input:** embedding model $\psi_\theta^0$, meta-training set $\mathcal{T} = \{S_t, Q_t\}_{t=1}^T$, number of classes in a task $K$
**Initialization:** sample tasks from $\mathcal{T}$ to initialize clusters $G = \{g_j\}_{j=1}^J$,

  **While** $|G|$ has not converged:
     $N_v = 1$ for each $g_v \in G$
     **For** $(S, Q) \in \mathcal{T}$:
        Match global clusters $V = \{v_q\}_{q=1}^K$ via (11) for task $(X, Y) = S \cup Q$
        **If** $V$ has $K$ unique clusters
           Update cluster $v$ for each $v \in V$ via (12)
     $G \leftarrow \{g_v | g_v \in G, N_v \geq \text{threshold in}(13)\}$
**Return** $G$

---

samples $\{x_i\}_{i=1}^I$ sharing the same local label within a task to a single global cluster as follows,

$$v^* = \underset{v}{\operatorname{argmin}} \left\| \frac{1}{I} \sum_{i=1}^I \psi_\theta^0(x_i) - g_v \right\|^2. \tag{11}$$

We apply (11) to each class of samples in a task, matching $K$ clusters in total. For simplicity, we discard tasks in which multiple local classes map to the same cluster. Otherwise, we update each matched cluster $g_{v^*}$ and sample counts $N_{v^*}$ with

$$g_{v^*} = \frac{N_{v^*} g_{v^*} + \sum_{i=1}^I (\psi_\theta^0(x_i))}{N_{v^*} + I} \quad \text{and} \quad N_{v^*} = N_{v^*} + I, \tag{12}$$

**Pruning Step.** We present a simple pruning strategy to regulate the number of clusters. Under the mild assumption that each cluster is equally likely to appear in a task, a cluster $v$ is sampled with probability $p = \frac{K}{J}$ for each $K$-way classification task. The number of samples $N_v$ assigned to cluster $v$ thus follows a binomial distribution with $N_v \propto B(T, p)$ where we recall $T$ as the size of the meta-training set. We remove any cluster below the threshold

$$N_v < \bar{N}_v - q\sqrt{\operatorname{Var}(N_v)} \tag{13}$$

where $\bar{N}_v$ is the mean of $N_v$, $\operatorname{Var}(N_v)$ the variance, and $q$ a hyper-parameter controlling the the aggressiveness of the pruning process.

Alg. 2 initializes a large number of clusters and populates the centroids with mean class embeddings computed from random tasks in $\mathcal{T}$. For $J$ initial clusters, $\lceil \frac{J}{K} \rceil$ tasks are needed since each task contributes $K$ embeddings. The algorithm alternates between clustering and pruning to refine the centroids and estimate the number of clusters. When the number of existing clusters no longer changes, the algorithm terminates and returns the current centroids $G$. Given the clusters $G$, all samples from the meta-training set can be assigned global labels.

We comment on two important points about Alg. 2. Firstly, the two hyperparameters, initial cluster count and pruning threshold, are only necessary when global labels are unavailable, since they determine the appropriate number of clusters. In contrast, access to global labels implies that the number of clusters and even the number of samples for each cluster is known. This further shows how global labels could oversimplify meta-learning as discussed earlier. Secondly, Alg. 2 differs from the classical $K$-mean algorithm [11] by exploiting local information to guide the clustering process, while $K$-mean algorithm is fully unsupervised. We will show empirically that enforcing local constraints is necessary for learning robust models.

**Meta-Learning with Inferred Labels.** After obtaining the inferred labels, we may apply a wide range of meta-learning algorithms (such as [27, 28] that exploit global labels) to obtain the final model. To highlight the efficacy of pre-training and the robustness of the proposed labeling algorithm, we choose [23] without additional fine-tuning in this work.

## 5 Experiments

We evaluate our proposed method on several benchmark datasets, including ImageNet variants, CIFAR variants, and a subset of MetaDataset [24]. As discussed earlier, we adopt independent

annotation with local labels only for all experiments. Model details and hyperparameter values are included in the supplementary material. Due to space constraint, CIFAR experiments and some ablation studies are also presented in the supplementary material.

The experiments aim to address the following questions: **1)** How does MeLa compare to existing algorithms? **2)** How does pre-training affect the performance of meta-learning algorithms? **3)** Does MeLa learn meaningful clusters?

**Experiments on ImageNet Variants.** We compare MeLa to a representative set of meta-learning algorithms on *mini*IMAGENET [25] and *tiered*IMAGENET [17]. For completeness, we include methods requiring access to global labels. However, we emphasize that these methods are not directly comparable to MeLa, since access to global labels provide significantly more information to meta-learners as discussed previously. These methods are intended to demonstrate the effect of pre-training on generalization performance. We also include self-supervised methods in the comparison.

Table 1: Classification accuracy of meta-learning models on *mini*IMAGENET and *tiered*IMAGENET.

|  | *mini*IMAGENET | | *tiered*IMAGENET | |
| --- | --- | --- | --- | --- |
|  | 1-shot | 5-shot | 1-shot | 5-shot |
| Global Labels | | | | |
| LEO [19] | $61.7 \pm 0.7$ | $77.6 \pm 0.4$ | $66.3 \pm 0.7$ | $81.4 \pm 0.6$ |
| RFS [23] | $62.0 \pm 0.4$ | $79.6 \pm 0.3$ | $69.4 \pm 0.5$ | $84.4 \pm 0.3$ |
| FEAT [28] | $66.7 \pm 0.2$ | $82.0 \pm 0.1$ | $70.8 \pm 0.2$ | $84.8 \pm 0.2$ |
| FRN [27] | $66.4 \pm 0.2$ | $82.8 \pm 0.1$ | $71.2 \pm 0.2$ | $86.0 \pm 0.2$ |
| Local Labels | | | | |
| MAML [5] | $48.7 \pm 1.8$ | $63.1 \pm 0.9$ | $51.7 \pm 1.8$ | $70.3 \pm 0.8$ |
| ProtoNet [20] | $49.4 \pm 0.8$ | $68.2 \pm 0.7$ | $53.3 \pm 0.9$ | $72.7 \pm 0.7$ |
| R2D2 [1] | $51.9 \pm 0.2$ | $68.7 \pm 0.2$ | - | - |
| MetaOptNet [10] | $\mathbf{62.6 \pm 0.6}$ | $78.6 \pm 0.5$ | $66.0 \pm 0.7$ | $81.5 \pm 0.6$ |
| Shot-free [16] | $59.0 \pm \text{n/a}$ | $77.6 \pm \text{n/a}$ | $63.5 \pm \text{n/a}$ | $82.6 \pm \text{n/a}$ |
| Initial Embedding (Eq. (3)) | $60.2 \pm 0.3$ | $75.6 \pm 0.5$ | $64.3 \pm 0.5$ | $78.9 \pm 0.4$ |
| MeLa (ours) | $\mathbf{62.0 \pm 0.4}$ | $\mathbf{79.6 \pm 0.3}$ | $\mathbf{69.1 \pm 0.5}$ | $\mathbf{84.2 \pm 0.3}$ |
| No Labels (Self-Supervised) | | | | |
| MoCo [8] (reported in [23]) | $54.2 \pm 0.9$ | $73.0 \pm 0.6$ | - | - |
| CMC [22] (reported in [23]) | $56.1 \pm 0.9$ | $73.9 \pm 0.7$ | - | - |

Similar to Fig. 2, Tab. 1 clearly shows the significant advantages of having access to global labels: all methods exploiting pre-training achieves noticeably higher generalization performance compared to methods without global labels. The results are consistent with our theoretical analysis that conditional meta-learning is more advantageous. Further, we observe that global labels not only enable pre-training but also flexible task sampling, including practical heuristics such as sampling more shots and more classes per task during meta-training [10, 27, 28]. All of the above contribute to generalization performance, and it is clear that global labels should be used when available.

In our experiment setting of "local labels" only, MeLa outperforms all baselines in three out of four settings, and obtains performance comparable to [10] in the remaining one. We highlight the comparison between the initial embedding $\psi_\theta^0$ obtained via (3) and the final embedding $\psi_\theta^*$ obtained by MeLa as they share identical experimental setups. It is thus easy to attribute the performance improvements to the proposed algorithm and the effect of pre-training. In particular, MeLa improves the average test performance by over 2% in *mini*IMAGENET and over 4% in *tiered*IMAGENET. In addition, we observe that MeLa obtains performance comparable to RFS [23], which is the oracle setting (i.e. access to ground truth global labels) for MeLa.

While FEAT and FRN outperforms MeLa in Tab. 1, we reiterate that methods exploiting global labels are not directly comparable, as discussed earlier. In addition, the performance of MeLa could be readily improved via more sophisticated data augmentation or fine-tuning. We explore one such variant for MeLa in Appendix B.2.

Table 2: The effects of initial embedding function on the labeling algorithm

| Dataset | *mini*IMAGENET | | *tiered*IMAGENET | |
| Replacement | Yes | No | Yes | No |
|---|---|---|---|---|
| Percentage of Tasks Clustered (%) | 100 | 98.6 | 99.9 | 89.5 |
| Clustering Acc (%) | 100 | 99.5 | 96.4 | 96.4 |
| 1-shot Acc (%) | $62.0 \pm 0.4$ | $61.8 \pm 0.5$ | $69.1 \pm 0.5$ | $69.1 \pm 0.5$ |
| 5-shot Acc (%) | $79.6 \pm 0.3$ | $79.4 \pm 0.4$ | $84.2 \pm 0.3$ | $84.1 \pm 0.3$ |

Table 3: Comparison between Alg. 2 and $K$-mean Clustering

| | *mini*IMAGENET | | | *tiered*IMAGENET | | |
| Cluster Alg. | Cluster Acc | 1-shot | 5-shot | Cluster Acc | 1-shot | 5-shot |
|---|---|---|---|---|---|---|
| Alg. 2 (MeLa) | 100 | $62.0 \pm 0.4$ | $79.6 \pm 0.3$ | 96.4 | $69.0 \pm 0.5$ | $84.1 \pm 0.3$ |
| $K$-mean | 84.9 | $60.7 \pm 0.5$ | $76.9 \pm 0.3$ | 28.2 | $64.8 \pm 0.6$ | $78.8 \pm 0.5$ |

In [23], the authors also adapted two state-of-the-art self-supervised methods MoCo [8] and CMC [22] for FSL. In the adapted methods, labels are not explicitly provided as input, but used to construct contrasting samples for learning self-supervised embedding. MeLa also outperforms them in Tab. 1.

**Robustness of the Labeling Algorithm.** A potentially trivial solution for clustering samples without any learning is by looking for identical samples: an identical sample appearing in multiple tasks would allow several local classes to be assigned to the same cluster. To demonstrate that Alg. 2 is not trivially matching identical samples across task, we introduce a more challenging experiment setting: each sample only appears once in the meta-training set. This implementation constructs the meta-training set by sampling from a flat dataset *without replacement*[3]. Consequently, Alg. 2 must only rely on the initial embedding function $\psi_\theta^0$ for estimating sample similarity. We evaluate MeLa under this setting and report the results in Tab. 2.

In Tab. 2, clustering accuracy is computed by assigning the most frequent ground truth label in each cluster as the desired target. In addition, percentage of tasks clustered refers to the tasks that map to $K$ unique clusters by Alg. 2. The clustered tasks satisfy both constraints imposed by local labels and are used for pre-training.

The results suggest that Alg. 2 is robust in inferring accurate global labels, even when samples do not repeat across tasks. The no-replacement setting also has negligible impact on test performance. In particular, we note that *mini*IMAGENET is particularly challenging under the new setting, with only 384 tasks in the meta-training set. In contrast, the typical sample-with-replacement setting has access to unlimited number of tasks for training.

The high clustering accuracy implies that the meta-distribution underlying the meta-training set is near perfectly recovered. MeLa thus effectively bridge the gap between experiment settings with and without access to global labels respectively. Using the inferred labels, we may apply a wide range of meta-learning algorithms to obtain the final model. We may also adopt flexible task sampling, such as sampling more shots and classes per task [10, 27, 28], for better generalization performance.

**The Importance of Local Constraints.** The clustering process enforces consistent assignment of task samples given their local labels. To understand the importance of enforcing these constraints, we consider an ablation study where Alg. 2 is replaced with standard $K$-mean algorithm, while other components of MeLa remain unchanged. $K$-mean algorithm is fully unsupervised and ignores any local constraints. We initialize the $K$-mean algorithm with 64 clusters for *mini*IMAGENET and 351 clusters for *tiered*IMAGENET, the actual numbers of classes in respective datasets.

Tab. 3 indicates that enforcing local constraints is critical in accurately inferring the global labels, as measured by clustering accuracy. In addition, lower clustering accuracy directly translates to lower test accuracy during meta-testing, suggesting that sensible task merging is an important prerequisite for feature pre-training. In particular, test accuracy drops by over 5% for *tiered*IMAGENET, when $K$-mean algorithm ignores local task constraints.

---

[3]For instance, *mini*IMAGENET (38400 training samples) can be split into 384 tasks of 100 samples in this setting.

Table 4: Meta-learning performance on MetaDataset subset (Aircraft, CUB and VGG)

| Algorithm | 1-shot | 5-shot |
|---|---|---|
| FEAT (no pre-train) | $60.9 \pm 0.7$ | $75.0 \pm 0.5$ |
| FRN (no pre-train) | $63.1 \pm 0.7$ | $79.7 \pm 0.5$ |
| Initial Embedding Eq. (3) | $61.9 \pm 0.5$ | $78.6 \pm 0.4$ |
| MeLa | $\mathbf{66.3 \pm 0.5}$ | $\mathbf{82.4 \pm 0.3}$ |

Between the two local constraints, we note that (11) is more important for accurately inferring global labels. Specifically, the clustering step improves accuracy by averaging the votes from all samples sharing the same local label in a task. On the other hand, the constraint on matching $K$ unique clusters are satisfied by almost all tasks empirically (see Tab. 2).

**Experiment on MetaDataset.** We further evaluate MeLa on MetaDataset [24], a collection of independently annotated datasets designed for meta-learning. MetaDataset presents a more challenging setting by including more diverse samples from different image domains.

We choose Aircraft, CUB and VGG flower datasets from MetaDataset for the experiment. The chosen datasets are all intended for fine-grained classification in aircraft models, bird species and flower species respectively. We compare MeLa against FEAT and FRN, two state-of-the-art methods. Since only local labels are used, all models are trained without pre-training. For meta-training, all models are trained on tasks sampled from the three datasets. For meta-testing, we sample 1500 tasks from each dataset and report the average accuracy. Test accuracy on individual datasets are included in Appendix B.4.

Tab. 4 shows that MeLa outperforms both FEAT and FRN when pre-training becomes inapplicable in the "local label" setting. In particular, FEAT performed relatively poorly since it is not designed to train from random initialization. In addition, MeLa improves upon the initial embedding by over 4%, similar to the performance gain observed in *tiered*IMAGENET. The results further validate the efficacy of pre-training even when the labels are inferred, consistent with our theoretical analysis. Lastly, we report that the clustering accuracy in this experiment is $79.3\%$, with a $94.7$ percent of the tasks clustered. The results suggest that MeLa's performance is robust to noise in the inferred labels.

# 6 Discussion

In this paper, we studied the effect of pre-training and the critical role of global labels in meta-learning. We showed that pre-training closely relates to meta-learning as a loss upper bound, and induces a conditional meta-learning formulation that explains the improved empirical performance. The effect of pre-training is consistently demonstrated in existing results and our experiments. The connection between meta-learning and pre-training opens up new opportunities of directly applying existing techniques from supervised learning towards meta-learning. For instance, model distillation [23] has been shown to further improve pre-training performance.

We also proposed a practical framework to infer global labels, for settings when they are unavailable. We demonstrate that MeLa is robust and accurate in inferring global labels, and achieves generalization performance competitive with state-of-the-art methods. In the ablation studies, we observed that meta-learning methods implicitly learn to cluster samples across tasks, even when samples do not repeat. In addition, explicitly enforcing the local constraints is critical for accurately inferring global labels and learning robust few-shot models.

**Limitations and Future Work.** We close by discussing some limitations of our work and directions of future research. In this paper, we focused on understanding the connection between pre-training and meta-learning, and evaluated MeLa on benchmarks with globally disjoint classes. In the future, we intend to extend our method to settings where global labels are ill-defined, such as when classes are overlapping or hierarchical. One possible approach is to assign soft labels or multiple labels to samples, thus capturing more complex relationship between classes.

## Broader Impact

Meta-learning aims to construct learning models capable of learning from experiences, Its intended users are thus primarily non-experts who require automated machine learning services, which may occur in a wide range of potential applications such as recommender systems and autoML. The authors do not expect the work to address or introduce any societal or ethical issues.

## Acknowledgments and Disclosure of Funding

The authors would like to thank the anonymous reviewers for their comments and suggestions. This work was supported in part by Career Development Fund (grant C210812045) from A*STAR Singapore, SAP SE, Royal Society (grant SPREM RGS\R1\201149) and Amazon Research Award (ARA).

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
