# Supplementary Material: The Role of Global Labels in Few-Shot Classification and How to Infer Them

The supplementary material is organized as follows:

- Appendix A contains the proofs accompanying our theoretical analysis.
- Appendix B presents additional experiment results.
- Appendix C details experimental setups, model architecture and hyperparameter values.

## A    Proofs

Let $\mathcal{T} = \{(S_t, Q_t)\}_{t=1}^T$ be a meta-training set where all tasks are annotated with global labels. We denote the collection of query sets as $\mathcal{Q} = \{Q_t\}_{t=1}^T = \{(X_t, Y_t)\}_{t=1}^T$ where we write $Q_t = (X_t, Y_t)$ as a tuple of query input samples $X_t = \{x_{jt}\}_{j=1}^{n_q}$ and their corresponding labels $Y_t = \{y_{jt}\}_{j=1}^{n_q}$. For simplicity, we assume that the query sets are disjoint, namely $Q_t \cap Q_{t'} = \emptyset$ for any $t \neq t'$. We merge all query sets into a flat dataset $D(\mathcal{Q}) = \{(x_i, y_i)\}_{i=1}^N = \cup_{t=1}^T Q_t$, with $N = n_q T$.

**Proposition A.1.** *With the notation and assumptions introduced above, let $C$ be the total number of classes in $D(\mathcal{Q})$, and $W \in \mathbb{R}^{C \times m}$ the global classifier. Denote by $W[Y]$ the sub-matrix with rows indexed by the sorted unique values from $Y$. Then, for any embedding $\psi_\theta : \mathcal{X} \to \mathbb{R}^m$*

$$\mathbb{E}_{(X,Y) \in \mathcal{Q}}\Big[ \mathcal{L}_{\mathrm{ce}}\big(W[Y], (\psi_\theta(X), Y)\big)\Big] \leq \mathbb{E}_{(x,y) \in D(\mathcal{Q})} \left[\ell_{\mathrm{ce}}(W\psi_\theta(x), y)\right], \qquad \text{(A.1)}$$

*Proof.* For a dataset $D$, let $\pi(D)$ be the set of class labels from $D$.

$$\mathbb{E}_{(x,y) \in D} \left[\ell_{\mathrm{ce}}(W\psi_\theta(x), y)\right] = \frac{1}{N} \sum_{(x,y) \in D} -\log \frac{\exp(W[y]\psi_\theta(x))}{\sum_{y' \in \pi(D(\mathcal{Q}))} \exp(W[y']\psi_\theta(x))} \qquad \text{(A.2)}$$

$$= \frac{1}{N} \sum_{Q \in \mathcal{Q}} \sum_{(x,y) \in Q} -\log \frac{\exp(W[y]\psi_\theta(x))}{\sum_{y' \in \pi(D(\mathcal{Q}))} \exp(W[y']\psi_\theta(x))} \qquad \text{(A.3)}$$

(A.3) rewrites the cross-entropy loss by enumerating over $\mathcal{Q}$. We observe that $\mathcal{Q}$ and $D(\mathcal{Q})$ share the same collection of samples, since all query sets are disjoint.

$$\text{(A.3)} \geq \frac{1}{T} \sum_{Q \in \mathcal{Q}} \frac{1}{n_q} \sum_{(x,y) \in Q} -\log \frac{\exp(W[y]\psi_\theta(x))}{\sum_{y' \in \pi(Q)} \exp(W[y']\psi_\theta(x))} \qquad \text{(A.4)}$$

$$= \frac{1}{T} \sum_{(X,Y) \in \mathcal{Q}} \Big[ \mathcal{L}_{\mathrm{ce}}(W[Y], (\psi_\theta(X), Y)) \Big] \qquad \text{(A.5)}$$

$$= \mathbb{E}_{(X,Y) \in \mathcal{Q}} \Big[ \mathcal{L}_{\mathrm{ce}}(W[Y], (\psi_\theta(X), Y)) \Big] \qquad \text{(A.6)}$$

In (A.4), the inequality is formed because the denominator $\sum_{y' \in \pi(D(\mathcal{Q}))} \exp(W[y']^\top \psi_\theta(x))$ is replaced with smaller values by summing over a smaller number of classes from $\pi(Q)$. Lastly, we rewrites the equation as the expectation over tasks in $\mathcal{Q}$.

Taking (A.2) and (A.6) yields

$$\mathbb{E}_{(X,Y) \in \mathcal{Q}}\Big[ \mathcal{L}_{\mathrm{ce}}\big(W[Y], (\psi_\theta(X), Y)\big)\Big] \leq \mathbb{E}_{(x,y) \in D(\mathcal{Q})} \left[\ell_{\mathrm{ce}}(W\psi_\theta(x), y)\right] \qquad \text{(A.7)}$$

$\square$

**Remark A.1.** *If $\mathbb{E}_{(x,y) \in D(\mathcal{Q})} \left[\ell_{\mathrm{ce}}(W\psi_\theta(x), y)\right] = 0$,*

$$\mathbb{E}_{(X,Y) \in \mathcal{Q}}\Big[ \mathcal{L}_{\mathrm{ce}}\big(W[Y], (\psi_\theta(X), Y)\big)\Big] = \mathbb{E}_{(x,y) \in D(\mathcal{Q})} \left[\ell_{\mathrm{ce}}(W\psi_\theta(x), y)\right] = 0.$$

*Proof.* If $\mathbb{E}_{(x,y)\in D(\mathcal{Q})}\left[\ell_{\mathrm{ce}}(W\psi_\theta(x), y)\right] = 0$, By Proposition 1,

$$\mathbb{E}_{(X,Y)\in\mathcal{Q}}\left[\mathcal{L}_{\mathrm{ce}}\big(W[Y], (\psi_\theta(X), Y)\big)\right] \leq 0$$

As cross-entropy loss $\mathcal{L}_{\mathrm{ce}}(\cdot) \geq 0$, we have $\mathbb{E}_{(X,Y)\in\mathcal{Q}}\left[\mathcal{L}_{\mathrm{ce}}\big(W[Y], (\psi_\theta(X), Y)\big)\right] = 0$ □

## B  Additional Experiments

### B.1  Impact of Pruning Threshold

In Algorithm 2, the pruning threshold is controlled by the hyper-parameter $q$. We investigate how different $q$ values affect the number of clusters estimated by the labeling algorithm and the corresponding test accuracy on *mini*IMAGENET and *tiered*IMAGENET.

Table 5: The effects of pruning threshold on test accuracy and the number of clusters.

| *mini*IMAGENET (64 classes) | | | | *tiered*IMAGENET (351 classes) | | | |
|---|---|---|---|---|---|---|---|
| $q$ | No. Clusters | 1-shot(%) | 5-shot(%) | $q$ | No. Clusters | 1-shot(%) | 5-shot(%) |
| 4.5 | 58 | $60.9 \pm 0.5$ | $78.5 \pm 0.4$ | 4 | 363 | $69.1 \pm 0.5$ | $84.2 \pm 0.3$ |
| 5.5 | 58 | $60.9 \pm 0.5$ | $78.5 \pm 0.4$ | 4.5 | 427 | $68.5 \pm 0.4$ | $83.6 \pm 0.3$ |
| 6.5 | 64 | $62.0 \pm 0.4$ | $79.6 \pm 0.3$ | 5.5 | 752 | $68.4 \pm 0.4$ | $83.5 \pm 0.3$ |

The results suggest that MeLa is robust to a wide range of $q$ and obtains similar performance for different $q$ values. With appropriate $q$ values, the number of clusters estimated for the two datasets are very close to the actual number of global classes. For *mini*IMAGENET, the labeling algorithm could recover exactly 64 classes. While it is possible to replace $q$ with directly guessing the number of clusters in Algorithm 2, we note that tuning for $q$ is more convenient since appropriate $q$ values appear to concentrate within a much narrower range, compared to the possible numbers of clusters.

### B.2  Experiment on MeLa Variant

MeLa is compatible with different meta-learning algorithms. In this experiment, we demonstrate that we could further exploit the performance gains from pre-training by leveraging S2M2 [12], which combines additional data augmentation during pre-training and fine-tuning after obtaining the inferred labels. In particular, S2M2 introduces two additional augmentation techniques, including sample rotation and sample mix-up [30]. The pre-trained model is fine-tuned with meta-training tasks to obtain embeddings more suitable for meta-testing.

We compared MeLa (S2M2) with several recent meta-learning methods, including RFS [23], FEAT [28] and FRN [27] in Table 6.

Table 6: Classification accuracy of meta-learning models on *mini*IMAGENET and *tiered*IMAGENET.

| | *mini*IMAGENET | | *tiered*IMAGENET | |
|---|---|---|---|---|
| | 1-shot | 5-shot | 1-shot | 5-shot |
| Global Labels | | | | |
| RFS [23] | $62.0 \pm 0.4$ | $79.6 \pm 0.3$ | $69.4 \pm 0.5$ | $84.4 \pm 0.3$ |
| FEAT [28] | $66.7 \pm 0.2$ | $82.0 \pm 0.1$ | $70.8 \pm 0.2$ | $84.8 \pm 0.2$ |
| FRN [27] | $66.4 \pm 0.2$ | $82.8 \pm 0.1$ | $71.2 \pm 0.2$ | $86.0 \pm 0.2$ |
| Local Labels | | | | |
| FRN (no pre-training) | $63.0 \pm 0.2$ | $78.01 \pm 0.2$ | - | - |
| MeLa (S2M2) | $65.8 \pm 0.4$ | $83.1 \pm 0.3$ | $70.6 \pm 0.5$ | $85.9 \pm 0.3$ |

Despite not having access to global labels, MeLa (S2M2) is highly competitive with FEAT and FRN, two state-of-the-art models that exploits global labels. In addition, the proposed method outperforms

RFS and FRN (no pre-training). The results further validate the efficacy of pre-training and the positive contribution from additional augmentation.

## B.3 Experiment on CIFAR Variants

In this section we present additional experiments on CIFAR-FS and CIFAR-100 datasets.

The CIFAR-FS dataset [1] is derived from the original CIFAR-100 dataset by randomly splitting 100 classes into 64, 16 and 20 classes for training, validation, and testing, respectively. The FC100 dataset [13] is also constructed from CIFAR-100 dataset with the classes split in a way similar to *tiered*IMAGENET. The exact splits used in our experiments are identical to [23]. We evaluate MeLa on both CIFAR-FS and FC100 in 5-way-1-shot and 5-way-5-shot settings.

Table 7: Comparison on CIFAR-FS and FC100 benchmarks

| | Accuracy (%) | | | |
| --- | --- | --- | --- | --- |
| | CIFAR-FS | | FC100 | |
| | 1-shot | 5-shot | 1-shot | 5-shot |
| MAML [5] | $58.9 \pm 1.9$ | $71.5 \pm 1.0$ | - | - |
| R2D2 [1] | $65.3 \pm 0.2$ | $79.4 \pm 0.1$ | - | - |
| TADAM [13] | - | - | $40.1 \pm 0.4$ | $56.1 \pm 0.4$ |
| Shot-free [16] | $69.2 \pm \text{n/a}$ | $84.7 \pm \text{n/a}$ | - | - |
| ProtoNet [20] | $72.2 \pm 0.7$ | $83.5 \pm 0.5$ | $37.5 \pm 0.6$ | $52.5 \pm 0.6$ |
| MetaOptNet [10] | $\mathbf{72.6 \pm 0.7}$ | $84.3 \pm 0.5$ | $41.1 \pm 0.6$ | $55.5 \pm 0.6$ |
| MeLa (Ours) | $71.4 \pm 0.5$ | $\mathbf{85.6 \pm 0.4}$ | $\mathbf{44.0 \pm 0.5}$ | $\mathbf{59.5 \pm 0.5}$ |
| RFS [23] | $71.6 \pm 0.5$ | $85.7 \pm 0.4$ | $44.4 \pm 0.5$ | $60.0 \pm 0.5$ |

Table 7 suggests that MeLa obtains test performance comparable to RFS, which is the oracle setting. This further validates that global labels may not be necessary as input, and that our proposed labeling algorithm is effective in inferring meaningful global labels across tasks. In addition, MeLa outperforms other meta-learning baselines in 3 out of 4 settings, and is only slightly worse than MetaOptNet in the remaining setting. While the high-dimensional embedding used by MetaOptNet (16000 dimensions vs 640 in ours) may be advantageous for some scenarios (e.g. CIFAR-FS 1-shot setting), they are potentially difficult to scale to larger tasks and pre-training still produces more robust embedding overall.

## B.4 Experiment on MetaDataset

We compare MeLa against the initial embeddings learned via Eq. (3), FEAT and FRN for fine-grained classification. Specifically, all models are trained on tasks sampled from from Aircraft, CUB and VGG flower. Since only local labels are used, all models are trained without pre-training. For meta-testing, we sample 1500 tasks from each constituent dataset and report the test accuracy for each dataset below.

Table 8: Test Accuracy on a subset of MetaDataset (Aircraft, CUB, VGG Flower). A single model is trained for each method over all tasks.

| | Aircraft | | CUB | | VGG Flower | |
| --- | --- | --- | --- | --- | --- | --- |
| | 1-shot | 5-shot | 1-shot | 5-shot | 1-shot | 5-shot |
| FEAT | $61.7 \pm 0.6$ | $75.8 \pm 0.5$ | $59.6 \pm 0.6$ | $73.1 \pm 0.5$ | $62.9 \pm 0.6$ | $76.0 \pm 0.4$ |
| FRN | $60.7 \pm 0.7$ | $77.6 \pm 0.5$ | $\mathbf{61.9 \pm 0.7}$ | $\mathbf{77.7 \pm 0.5}$ | $65.2 \pm 0.6$ | $81.2 \pm 0.5$ |
| Eq. (3) | $67.7 \pm 0.6$ | $82.8 \pm 0.4$ | $53.8 \pm 0.5$ | $69.2 \pm 0.5$ | $65.4 \pm 0.5$ | $83.3 \pm 0.3$ |
| MeLa | $\mathbf{69.2 \pm 0.5}$ | $\mathbf{84.3 \pm 0.4}$ | $61.0 \pm 0.5$ | $\mathbf{77.2 \pm 0.4}$ | $\mathbf{69.4 \pm 0.5}$ | $\mathbf{86.0 \pm 0.3}$ |

# C   Model and Experimental Setups

We provide additional details on the model architecture, experiment setups, and hyperparameter choices. We performed only limited model tuning, as it is not the focus on the work.

## C.1   Model Architecture

We use a ResNet-12 architecture for all our experiments. The architecture strikes a good balance between model complexity and performance, and is one of the most commonly adopted architecture in existing works [e.g. 10, 13, 16, 23]. Specifically, we adopt the default architecture from the official implementation[4] of [23]. The model's penultimate layer is averaged and outputs an embedding $\psi_\theta(x) \in \mathbb{R}^{640}$.

## C.2   Experiment Setup

The initial embedding function $\psi_\theta^0$ can be trained on either 1-shot or 5-shot setting with minimal impact on the quality of $\psi_\theta^*$. We choose the latter in our experiments. To ensure fair comparison, we follow the existing convention and use 15 samples per class for query sets $Q$.

For all experiments, we adopt an initial learning rate of 0.05. The learning rate is decayed by a factor of 0.1 twice for all datasets. All models are trained using a SGD optimizer with a momentum of 0.9 and a weight decay of $5 \times 10^{-4}$.

Table 9 reports hyperparameter values used in our experiments. Datasets CIFAR-FS and FC100 share the same values and are reported under "CIFAR".

Table 9: Hyperparameter values used in the experiments

| | | VALUES | | |
|---|---|---|---|---|
| SYMBOL | DESCRIPTION | *mini*IMAGENET | *tiered*IMAGENET | CIFAR |
| $\lambda_1$ IN (4) | REGULARIZER FOR RIDGE REGRESSION | $10^{-3}$ | $10^{-3}$ | $10^{-3}$ |
| $\lambda_2$ IN (7) | REGULARIZER FOR LOGISTIC REGRESSION | 1 | 1 | 1 |
| $J$ | INITIAL NUMBER OF CLUSTERS | 300 | 3000 | 300 |
| $q$ | PRUNING PARAMETER IN (13) | 6.5 | 4.5 | 4.5 |

## C.3   Meta-Testing

For all of our models, we use

$$w_{\text{ce}}(Z) = \underset{W}{\arg\min} \, \mathcal{L}_{\text{ce}}(W, Z) + \lambda_2 \|W\|^2 \tag{C.1}$$

as the base learner for meta-testing. (C.1) is implemented by scikit-learn[5] and identical to the one used in [23]. We observe empirically that this base learner outperforms other common choices such as ProtoNet [20] or SVM [10].

## C.4   Computational Requirements

All experiments are runnable on a commodity desktop PC with a single Nvidia 2080 Ti and 48GB of RAM. MeLa takes about 3 hours to train for *mini*IMAGENET and about 6 hours to train for *tiered*IMAGENET. CIFAR-FS and FC100 both take about 1.5 hours for training.

The computational complexity of MeLa is similar to other methods that exploits pre-training. In addition, our choice of applying ridge regression for learning the initial embedding function is computationally efficient and fast to train.

---

[4]https://github.com/WangYueFt/rfs
[5]https://scikit-learn.org/stable/