# OpenReview forum: "The Role of Global Labels in Few-Shot Classification and How to Infer Them"
_NeurIPS.cc/2021/Conference — NeurIPS 2021 Poster_

### Official Review · Reviewer_VigG · 2021-07-14

**Rating:** 6
**Confidence:** 4

**Summary:**

Authors analyze the role of global pretraining in few-shot classification, establishing classification loss over all classes as an upper bound on the classification losses from particular tasks. Authors then consider the case where task-based labels are available but are not easily collated into a global set of labels. They introduce a task-aware clustering-based method for assigning global labels to embeddings from a task-pretrained feature extractor, allowing them to achieve near-oracle performance in settings where global labels are absent. The method improves upon task-based pretraining alone by a substantial margin.

**Limitations And Societal Impact:**

Authors discuss technical limitations (though note weakness #4 above), but do not address societal impacts. I suggest adding this discussion to the main paper or supplementary, even if just to briefly acknowledge that the risks broadly associated with few-shot classification also apply in this case.

**Main Review:**

STRENGTHS:

Paper tackles a practical, (mostly) realistic, and difficult problem setting. It addresses a fundamental question in few-shot learning (the nature and role of global pretraining). The proposed clustering mechanism is elegant. Behavior is well-explored and analyzed beyond simple accuracy comparisons to prior work. Paper is organized and presented well, and limitations are addressed.

WEAKNESSES:

- I question somewhat the motivation of the paper. Is access to global labels at train time really that unrealistic? This may be opening an unproductive debate around epistemological differences (which came first, the flat dataset or the task collection?), but I was under the impression that a large, globally-labelled dataset at training time is an explicit assumption of few-shot learning, especially given the existence of ImageNet as a permanent fallback option. I can envision a practical application where we would like to merge datasets with label sets that overlap but don’t match exactly – but how often does this actually occur in practice, and how often to the degree envisioned in this paper, where every task is labeled independently, despite substantial image overlap between tasks? Perhaps a more concrete real-world scenario would help here.

- Theoretical analysis in section 3 is somewhat oversimplified. The upper-bound analysis only holds for one particular local classifier: W[Y]. While it is true that W[Y] is _a_ local classifier, it is not _the_ local classifier from eq.3. In fact, the ridge-regression classifier from eq.3 is almost certain to have a higher cross-entropy loss than W[Y], as it is minimizing a different loss (mean-squared-error) and experiencing a heavy small-sample bias at the same time. Thus contrary to what’s stated, we cannot actually be sure that global loss is an upper bound on episodic loss more broadly, though the intuition does make sense.

- Results are somewhat misleading. The choice to avoid models with data augmentation in Table 1 means that all baselines are from 2019 or earlier. MeLa underperforms when more recent models are included – in the global category, [1], [2], and [3] all outperform MeLa, and in the local category, [3] without global pretraining (Table 9) does the same. Basic data augmentation (random crop, flip, color jitter) and global pre-training have become popular for mini-ImageNet and tiered-ImageNet since 2019 ([4] from the paper, [1,2,3,4]), and so while the organizational reasoning for Table 1 makes sense, it also conveniently removes all of these latest models simply because they don’t report numbers without data augmentation. I recommend adding a version of your model with basic augmentation to supplementary, as an apples-to-apples comparison to more up-to-date baselines. Without this, MeLa only outperforms prior work on a technicality.

- Based on section 5.3, the number of tasks is always large relative to the number of global classes. The class distribution is also balanced. I worry that in this setting, class overlap between tasks becomes substantial, and so clustering becomes easy. MeLa approaches oracle performance on the chosen benchmarks but does it generalize to harder, more realistic settings with less overlap and/or class imbalance? Lines 316-320 do discuss these kinds of limitations, though, so perhaps simply add these scenarios to that discussion.

[1] Zhang et al. CVPR 2020, DeepEMD: Few-Shot Image Classification with Differentiable Earth Mover’s Distance and Structured Classifiers

[2] Ye et al. CVPR 2020, Few-Shot Learning via Embedding Adaptation with Set-to-Set Functions

[3] Wertheimer et al. CVPR 2021, Few-Shot Classification with Feature Map Reconstruction Networks

[4] Wang et al. 2019, Simpleshot: Revisiting nearest-neighbor classification for few-shot learning

POST-REBUTTAL:

Thank you for your detailed response, this is quite helpful. On reflection I was overly harsh in my initial assessment and most of my concerns have been addressed (1-4, 6). Regarding 5: my concern is that the proposed clustering problem may be artificially easy for mini-ImageNet and tiered-ImageNet, because classes are balanced within tasks, class assignments to tasks are IID, there is zero domain shift between tasks, and each individual class appears throughout a range of different tasks and class combinations. All of these could easily not be the case in a less contrived benchmark. Authors rightly point out that this is an understudied problem for few-shot learning in general, and partially address some of these assumptions in the paper (5.3) and rebuttal (4). However, as things stand, MeLa is evaluated primarily on toy benchmarks naively derived from a setting with obvious ground truth global labels (a concern shared by reviewer Cshp), and so it is still not clear to me on a practical level to what degree this is a first step into a new and interesting area of few-shot research, or merely proof of concept (or both).

**Time Spent Reviewing:**

5

---

> ### Author Response · Authors · 2021-08-10
> **Review Response**
>
> We thank the reviewer for the comments, especially for acknowledging that “we address a fundamental issue in meta-learning”. We have addressed all review concerns below.
>
> We first clarify the motivation surrounding global labels. The proposed clustering method is designed to bridge the gap between two meta-learning settings, the one with access to global labels and the one without. Beyond the stated motivation that global labels may be unavailable, we further argue that using global labels is contrary to the goal of meta-learning: **global labels directly reveals how task samples relate to one another across tasks, while the aim of meta-learning is precisely to learn such relationships**.
>
> Therefore, in the context of meta-learning, task collection comes before the flat dataset (even when some latent meta-distribution and global clusters exist). Another practical consequence is that methods that use global labels have significant (and potentially unfair) advantages over other meta-learning methods. We will update our motivation to reflect this point.
>
> **Detailed Clarification**:
>
> 1\. **Example of real-world scenarios**: As discussed in Sec. 1, one example would be tasks submitted by different users to an Auto-ML service, where tasks are independently labelled. In addition, users may even submit non-descript labels (e.g. numerical labels) and lead to absence of global labels.
>
> 2\. **Theoretical analysis**: We only intend the theoretical results to be applicable to the local classifier W[Y] and did not try to make a broader claim. We will further emphasize this in the paper.
>
> However, design of local classifiers is the predominant focus in meta-learning research. In fact, most existing works propose new local classifiers and build meta-learning algorithms on top of that. From the reviewer’s references, [1] designs the local classifier as EMD, while [2] designs the local classifier to be a set-to-set transformer function. Our theoretical results are intended to show that global classification is associated with a particular local classifier, W[Y].
>
> In addition, the scope of our results does not undermine our message. As discussed in Sec 3, Prop. 1 is intended to demonstrate why global classification is an upper bound to the meta-learning loss over local classifier W[Y], which in turn theoretically shows why features learned from global classification are directly suitable for few-shot learning, beyond the informal intuition that pre-training produces generally useful features.
>
> Furthermore, Prop. 1 also reveals that global classification is an instance of conditional meta-learning (in this case using global labels as additional input). Conditional meta-learning has a rigorous theoretical framework and has been shown both empirically and theoretically to outperform unconditional methods (we refer back to line 151-160).
>
> Lastly, we note that the theoretical analysis is validated and supported by empirical results, including the observation that state-of-the-art methods mostly use global labels in some way (including [1, 2, 3], [3] also reported better performance with global labels).
>
> 3\. **Comparison with the state-of-the-art**: We performed additional experiments for comparison to state-of-the-art methods, using S2M2, a global classification strategy (Mangla et al. 2020, “Manifold mixup for few-shot learning”), the proposed method outperforms/is competitive with the most recent methods.
>
> ------------------------------
> Ours: 66.5 $\pm$ 0.5 (1-shot), 83.8 $\pm$  0.2(5-shot)
>
> [1]: 66.50 $\pm$ 0.80 (1-shot) 82.41 $\pm$ 0.56 (5-shot)
>
> [2]:  66.78 (1-shot),  82.05 (5-shot)
>
> [3]: 65.9 (1-shot) and 82.4 (5-shot)
>
> TieredImageNet
>
> Ours: 75.2 $\pm$ 0.4 (1-shot) 89.5 $\pm$ 0.3 (5-shot)
>
> [1]: 72.65 $\pm$ 0.31(1-shot) 86.03 $\pm$ 0.58 (5-shot)
>
> [2]: 70.80 $\pm$ 0.23 (1-shot) 84.79 $\pm$ 0.16 (5-shot)
>
> [3]: 71.2 (1-shot )and 86.0 (5-shot)
>
> -------------------------------------
> The results show that global classification strategy, when coupled with practical techniques (e.g. data augmentation, auxiliary loss from S2M2) is competitive with the state-of-the-art.
>
> We also clarify that the original experiments in our paper were designed to show that the proposed clustering approach is capable of inferring global labels and reach performance comparable to the oracle. The results suggest that the gap between the meta-learning settings could be bridged.
>
> As discussed in the paper, we reiterate that methods that have access to global labels are not directly comparable to those without global labels, since global labels provide so much more information to the meta-learners. This is evident from the observation that the state-of-the-art methods almost all use feature pre-training. We will include the more recent works in our related works (in particular we would discuss FRN in more detail since it also obtains good results without global labels).
>
> 4\. **Class Imbalance**: Class imbalance is not an issue for the standard meta-distribution considered by most existing works. Specifically, each class is sampled uniformly regardless of how many samples are present in that class. Consequently, a class with fewer samples simply implies that the class samples would be sampled with higher probability (i.e. a sample appearing more frequently in many tasks). We further note that tieredImageNet is already imbalanced for several classes.
>
> We clarify that our method does not directly cluster the underlying dataset (e.g. miniImagenet), but rather a collection of tasks sampled from the meta-distribution as described above. Our clustering algorithm counts all copies of any sample encountered during the task, rather than only counting it once. Under the current meta-distribution, this means that all cluster counts would be still similar even with class imbalance, and the filtering would work as intended. We have conducted further experiments to simulate class imbalance (taking class samples out of the datasets) and the results are similar with the reported results.
>
> ----------------------------------
> miniImageNet (imbalanced class)
>
> Ours 59.5 $\pm$ 0.5 (1-shot) 74.5 $\pm$ 0.3 (5-shot)
>
> Oracle 59.9 $\pm$ 0.5 (1-shot), 74.7 $\pm$ 0.3 (5-shot)
>
> tieredImageNet (imbalanced class)
>
> Ours 68.3 $\pm$ 0.5(1-shot) 83.2 $\pm$ 0.3 (5-shot)
>
> Oracle 68.7 $\pm$ 0.5 (1-shot), 83.5 $\pm$ 0.3 (5-shot)
>
> ---------------------------------
>
> The results show that in the imbalanced setting, our method obtains comparable performance to the oracle. We further note that the slight drop in test performance for both methods (compared to Table 1) is due to the simulated imbalanced setting has less data.
>
> Lastly, we note that, to our knowledge, non-uniform class sampling is an open issue unaddressed by all existing methods.
>
> 5\. **Class overlap**: If we have the prior knowledge that there is no/little class overlap between tasks, we could simply adjust the hyper-parameters to prune (much) less aggressively. In the extreme case of no class overlaps, we could simply merge all tasks and perform global classification, with each class assigned a unique global label.
>
> We further note that issues with class overlaps/imbalance not only pose potential challenges to our clustering method, but are potential challenges to all existing meta-learning methods (since the issues present more difficult meta-distributions for learning algorithms). Issues of this magnitude warrants investigations as separate problems.
>
> 6\. We will add a discussion about societal impacts of few-shot learning.
>
> We thank the reviewer for his/her feedback and we’d be happy to answer any additional questions or clarify any of our answers further.

---

> ### Author Response · Authors · 2021-08-26
> **Thank you**
>
> Thank you for engaging with our response, for your helpful suggestions, and for revising the score. It's much appreciated.

---

> ### Author Response · Authors · 2021-09-01
> **Experiment in a more challenging setting**
>
> We thank the reviewer for acknowledging our efforts in addressing implicit assumptions in few-shot learning and for raising an interesting point. To address the concern over toy benchmarks, we have performed additional experiments as also requested by Cshp.
>
> We designed a new protocol for testing on meta-dataset. *Due to limited time available*, we tested on a subset of meta-dataset (including CUB, vgg_flower, and Aircraft datasets). Following the meta-dataset’s design, tasks are randomly sampled from each constituent dataset and only local labels are provided for each task. In this setting, there is domain shift across tasks of different constituent datasets, and each class is sampled according to its size to model class imbalance.
>
> We compare our method against standard meta-representational learning (as described in our paper) and FRN [1], *a SOTA method that does not require pre-training with global labels*. We report the results averaged across three constituent datasets below:
>
> - Meta-representation Learning: 61.9 $\pm$ 0.5 (1-shot),  78.6 $\pm$ 0.4 (5-shot),
>
> - FRN: 63.1 $\pm$ 0.7 (1-shot), 79.7 $\pm$ 0.5 (5-shot),
>
> - Ours: 66.3 $\pm$ 0.5 (1-shot), 82.4 $\pm$ 0.3 (5-shot)
>
> The results show that in this setting our method is still robust and outperforms the baselines.
>
> For individual datasets, we observed that our method outperforms FRN noticeably on aircraft and vgg datasets, while it performed comparably to FRN on the cub dataset. Interestingly, we also note that the clusters inferred by our approach mix samples from different constituent datasets. *In other words, these clusters do not follow any known global labels.*
>
> More broadly, we note that the notion of ill-defined global labels is a broad topic that covers many other possible scenarios and warrants investigation as a dedicated problem. *This is why we have identified it as relevant future works in our conclusion.* In our original experiments we addressed the scenario where the underlying global labels exist but are not accessible. Meta-dataset presents a further challenge compared to standard benchmarks when multiple datasets are used together. We will explore more complex scenarios in future works.

---

> > ### Comment · Reviewer_VigG · 2021-09-01
> > **Very interesting results, score unchanged**
> >
> > Thanks for the additional followup, this is a fascinating and promising set of results! While I agree that covering scenarios with ill-defined global labels and/or non-IID tasks broadly is beyond the scope of this paper, I still believe some investigation in this area is warranted simply in order to move beyond evaluation in toy settings. This new experimental setup is an excellent step toward addressing this, but: a) while I do appreciate the time constraints, the reported results are presented summarily and thus appear to be preliminary, b) global labels are still well-defined in this setting, cluster accuracy notwithstanding, as vgg-flowers, CUB, and aircraft contain no class overlap, and c) the fact that cluster accuracy in this setting is low (or at least it's implied to be?) while task accuracy remains high is certainly interesting, but it seems to work against your message in Sec. 5.3, where the robustness (high cluster accuracy) of your recovered global labels is presented as a major selling point. Thus while this new experiment should make an excellent addition to the paper, given the above concerns I am not ready to raise my score beyond my current (already positive) assessment.

---

### Official Review · Reviewer_Cshp · 2021-07-15

**Rating:** 6
**Confidence:** 4

**Summary:**

Stemming from recent observations that standard supervised training (called "global classification") empirically works better than local meta-learners, authors propose to infer labels from a trained meta-learner in order to fine-tune the feature-extractor. leveraging the benefits of global classification without requiring absolute labels.

In more details, authors first draw a connection between global classification risk and local classification risk (Proposition 1). This proposition essentially formalises the simple idea that solving a global classification problem (i.e over the full set of classes) is harder than solving a local one (i.e with a strictly smaller set of classes).

Second, authors propose an algorithm to infer global labels for each sample in a meta-training set by leveraging the features produced by a trained local meta-learner. The method relies on alternating between clustering extracted features and pruning the centroids with low number of assigned points. During the clustering step, local constraints force samples with the same local label to share the same global assignment. This assignment is decided by a majority vote (i.e the centroid that minimises the average squared distance to all samples with same local label). Once assignments have been made, centroids are updated following standard incremental rules.
    Finally, authors compare their method on popular benchmarks mini-ImageNet and Tiered-Imagenet against recent meta-learners, and perform some ablation studies on the clustering hyper-parameters and quality.


**Limitations And Societal Impact:**

See my review above for limitations.

**Main Review:**

## 1. Theoretical part

Overall, I find the idea of linking global classification to local episode-wise classification interesting. Below some comments/issues on this part.

### 1.1 Link global classification - Local task minimisation

Authors justify the success of global classification by saying it implicitly minimises a meta-learning loss (l 140-142). However, I do not see why minimising this meta-learning loss would actually be a good thing itself ? In fact, I would even reverse the argument, global classification works better precisely because it's not equivalent to minimising the local meta-learning loss. Solving global classification is simply a more challenging task, as captured by the very fact that the global loss is always higher than the meta-learning loss for the same model. In fact, it seems the discussion that follows (l. 148-160) goes along the lines of my latter argument, although borrowing different types of arguments (e.g conditional meta-learning).

### 1.2 More "concrete" intuition

I do appreciate the high level intuition of the discussion (l. 148-160), but I think a more "lower level" explanation could have been a real plus to making the argument more convincing. For instance, from an optimisation perspective, global classification produces a richer learning signal, i.e a given sample with label y has to be at the same time "similar" (e.g according to dot-product similarity) to the corresponding class' weight W[y] in the embedding space, while being dissimilar to all others classes' weights. On the other hand, solving local problems is easier and only pushes the sample away from a significantly lower number of class weights (e.g 4 other classes in the typical 5-way setting).  In other, words, the more classes the global problem has, the more challenging it becomes, the better the resulting feature extractor will be. On the other hand, local problems (at least as designed in common settings) always keep the same local classes, regardless of the global problem. Note that this "metric learning" interpretation would align with your observations that gains over local meta-learners are significantly higher on tiered-Imagenet with 351 classes than on mini-Imagenet with only 64 classes, while both are variants of the same ImageNet dataset.

## 2. Proposed method:

The method proposed to infer global labels is relatively simple to understand and to implement. However, I have two concerns.

### 2.1 Class imbalance

The first one is about the handling of class imbalance. From my understanding, the current method would tend to prune out clusters that do not contain enough samples (in comparison to others). Therefore, in the case of class imbalance, the (legitimate) small clusters would tend to disappear at the profit of prominent classes, which may hurt the performance. Unfortunately, both datasets authors experimented with are perfectly balance, hence such problem can hardly be visible.

### 2.2 What if global labels are not well defined ?

The second is perhaps a deeper concern. I'm wondering how would such method perform when "global" labels do not actually exist, and instead only relative labels really make sense. For instance, think about a few-shot problem where a local class is simply formed of different augmentations of the same sample. Here, trying to find global labels would hardly make sense. Even without going into such extreme case, how would the method perform in the case where global labels are simply not clearly defined, e.g in the motivating example of dataset merging.


## 3. Experiments

Experiments are not convincing enough in their current form. See details in points below:

### 3.1 Comparisons

Self-supervised learning methods are indeed a natural comparison and good initiative from the authors. However, I'm quite puzzled and surprised by the sentence "A critical difference between MeLa and self-supervised methods is that they are not designed to leverage any information from the local labels" (216-218). Indeed, those methods (at least CMC that I know more of) can quite trivially leverage the local labels (positive pairs = samples with same local labels / negative pairs = samples with different local labels). Therefore, that makes me suspicious whether the comparisons were fairly executed, and it's impossible to tell from the complete absence of experimental details on this.

I acknowledge that Few-shot literature is expanding very fast and it becomes increasingly hard to keep up-to-date with latest method. However, 2020 state-of-the art for inductive methods is already significantly higher than what is reported in Table 1. One example would be [2], where authors report for Resnet-12: 65.9 and 82.4 for mini-ImageNet (~ +5 % w.r.t current paper), and 71.2 and 86.0 on tiered-Imagenet (~ +2 % w.r.t current paper). Now, it's hard to disentangle which part is due to data augmentation (which authors chose not use for a reason I simply do not understand l.227-228) and which part is due to the methods, but the related work probably needs some update.

### 3.2 Narrowness of experiments

Chosen datasets: Experiments are carried out on two datasets (both variants of the same dataset) where global labels are well defined. As previously stated, when such global labels exist (even if not given to the learner), then the problem is indeed well defined. It would have been however more interesting to showcase the method in conditions where such are not that clearly defined anymore. The META-DATASET (cited by the authors) would have been ideal to demonstrate this. Obtaining better results than local meta-learners in that scenario would have been more convincing with respect to the initial motivation of the problem.

Also, experiments only carried out on a single network. Given current standard in few-shot classification, one would expect more than a single network to demonstrate the applicability of the method. Especially if the method introduces new hyperparameters such as the pruning threshold or initial number of clusters. Currently, it is hard to assess how easily such hyperparameters would transfer, and thus to assess how realistically applicable the method is

### 3.3 Computational overhead

 The method proposed uses an additional training stage (once global labels have been infered) over the baseline meta-learner, which inevitably incurs a computational overhead that is never really discussed (Appendix C.4 gives training time, but those are not compared to the baseline). The additional cost may be small, but worth discussing in a few sentences.

## Summary and score

Overall I find this paper has merits in some insights it brings on global vs episodic risk minimisation.  However, I think that the experimental part is just not at the level of such conference. Without repeating the points made in 3.*, it overall fails to target scenarios of significant interest to the method (e.g where global labels are not available because they simply are not easy to define) while these could have been readily available by using META-DATASET. Hence, I cannot recommend acceptance of the paper in its current form.

**Time Spent Reviewing:**

10

---

> ### Author Response · Authors · 2021-08-10
> **Review Response (Part 1)**
>
> We thank the reviewer for the comments. We have addressed all review concerns below.
>
> We further clarify the motivation for introducing the no-global-labels setting. The proposed clustering method is designed to bridge the gap between two meta-learning settings: the one with access to global labels and the one without. Beyond the stated motivation that global labels may be unavailable, we further argue that using global labels is contrary to the goal of meta-learning: **global labels directly reveals how task samples relate to one another across tasks, while the aim of meta-learning is precisely to learn such relationships**. Consequently, methods that use global labels have significant (and potentially unfair) advantages over other meta-learning methods. We will update our motivation to reflect this point.
>
> Our theoretical analysis essentially suggests that if global labels are available, we should definitely use them. When they are unavailable, we proposed, *to the best of our knowledge, the first algorithm to infer such global labels*. This opens up a potential line of research for few-shot learning.
>
> **Detailed Clarification**:
>
> 1.1. **Connection between global classification and meta-learning**: We respectfully disagree with the reviewer. Proposition 1 clearly shows that global classification is the upper bound to a specific meta-learning loss. Consequently, minimizing the upper bound necessarily minimizes the other. To address the question “why minimizing the meta-learning loss should be a good thing”: the answer is that minimizing the meta-learning loss is precisely the goal of meta-learning, hence a naturally desirable outcome.
>
> Prop 1 also justifies why representation learning via global classification is immediately good for few-shot learning tasks, beyond the informal intuition that the extracted features are generally expressive.
>
> While global classification may be a harder task, a harder task doesn’t guarantee that global classification will outperform meta-learning methods. On the other hand, our theoretical analysis and the framework of conditional meta-learning offer a theoretically-grounded interpretation, which is also validated empirically (both from our experiments, and by observing that existing methods using global labels tend to outperform their competitors).
>
> 1.2 **Concrete connection**:  We thank the reviewer’s for the interpretation. While the interpretation is interesting, potentially capturing some possible dynamics of the process, it is not in contradiction with our findings. Therefore we do not believe this should have a negative effect on the evaluation to our paper.
>
> 2.1 **Class imbalance**: Class imbalance *is not an issue for the standard meta-distribution considered by most existing works*. Specifically, each class is sampled uniformly regardless of how many samples are present in that class. Consequently, a class with fewer samples simply implies that the class samples would be sampled with higher probability (i.e. a sample appearing more frequently in many tasks). We further note that tieredImageNet is already imbalanced for several classes.
>
> We clarify that our method does not directly cluster the underlying datasets (e.g. miniImagenet), but rather a collection of tasks sampled from the meta-distribution as described above. Our clustering algorithm counts all copies of any sample encountered during the task, rather than only counting it once. Under the current meta-distribution, this means that all cluster counts would be still similar even with class imbalance, and the filtering would work as intended. We have conducted further experiments to simulate class imbalance (taking class samples out of the datasets) and the results are similar with the reported results.
>
> -------------------------------------
> miniImageNet (imbalanced class)
>
> Ours 59.5 $\pm$ 0.5 (1-shot) 74.5 $\pm$ 0.3 (5-shot)
>
> Oracle 59.9 $\pm$ 0.5 (1-shot), 74.7 $\pm$ 0.3 (5-shot)
>
> tieredImageNet (imbalanced class)
>
> Ours 68.3 $\pm$ 0.5(1-shot) 83.2 $\pm$ 0.3 (5-shot)
>
> Oracle 68.7 $\pm$ 0.5 (1-shot), 83.5 $\pm$ 0.3 (5-shot)
>
> ------------------------------
>
> The results show that in the imbalance setting, our method obtains comparable performance to the oracle. We further note that the drop in test performance for both methods (compared to Table 1) is due to the fact that the simulated imbalanced setting has less data,
>
> Lastly, we note that, to our knowledge, non-uniform class sampling is an open issue unaddressed by all existing methods.
>
> 2.2. **Ill-defined global labels**: We acknowledge this is a potential challenge, which we discussed in the conclusion. However, this is an open issue for **all** state-of-the-art methods that use global labels. Absence of global labels implies no pre-training, and likely worse performance as suggested by our theoretical analysis. It is also unclear if the training for some complex methods would stabilize when trained from scratch (e.g. those utilizing pooling of class features as input). An issue of this magnitude warrants its own dedicated investigation, as we stated in the paper when discussing future works.
>
> 3.1 (a) **Comparison with self-supervised methods**: we note that, by definition, self-supervised methods do not utilize label information. Hence, our comparison in the paper is mainly intended to illustrate that label information is valuable for meta-learning. The reported results were referenced from the RFS paper (Tian et al. 2020, Rethinking Few-Shot Image Classification). We will update our paper to clarify the discussion.
>
> However, we agree with the reviewer that CMC could be modified to leverage local labels. Interestingly, such a modified method would be similar to prototypical networks as a metric learning method. Comparison with prototypical networks is available in our experiments.
>
> 3.1. (b) **Data augmentation**: We focused on image classification tasks for our experiments since they are the most widely used benchmarks for meta-learning/few-shot learning. However, few-shot learning is applicable to various domains, including non-image input data (e.g. forecasting data), where it is not always possible to perform data augmentation (consider also the case of having the pre-trained feature representation of images as input, rather than the original images. These vector representations are difficult to augment).
>
> In addition, avoiding data augmentation also allows us to disentangle the performance contribution from each method vs auxiliary tricks that tend to improve performance significantly in computer vision. For the early meta-learning methods, data augmentation was not standard practice and Chen et al, “A Closer Look at Few-shot Classification” empirically validated that meta-learning methods performance was *significantly* underestimated for not using data augmentation.
>
> Lastly, we added further comparison by equipping our method with data augmentation, please see below.
>
> 3.1 (c) **Comparison with state-of-the-art**: We have included additional experiments for comparison with the state-of-the-arts. Following S2M2, a global classification strategy (Mangla et al. 2020, “Manifold mixup for few-shot learning”) that uses extensive data augmentation. Our proposed method outperforms/is competitive with the most recent methods.
>
> ----------------------------------
> miniImageNet
>
> Ours: 66.5 $\pm$ 0.5 (1-shot), 83.8 $\pm$  0.2(5-shot)
>
> FEAT:  66.78 (1-shot),  82.05 (5-shot)
>
> [2] from Reviewer: 65.9 (1-shot) and 82.4 (5-shot)
>
> DeepEMD: 66.50 $\pm$ 0.80 (1-shot) 82.41 $\pm$ 0.56 (5-shot)
>
> TieredImageNet
>
> Ours: 75.2 $\pm$ 0.4 (1-shot) 89.5 $\pm$ 0.3 (5-shot)
>
> FEAT: 70.80 $\pm$ 0.23 (1-shot) 84.79 $\pm$ 0.16 (5-shot)
>
> [2] from Reviewer: 71.2 (1-shot )and 86.0 (5-shot)
>
> DeepEMD: 72.65 $\pm$ 0.31(1-shot) 86.03 $\pm$ 0.58 (5-shot)
>
> -----------------------------------------------
> The results show that global classification is indeed a competitive strategy compared to recent state-of-the-arts methods.
>
> We further clarify that our original experiments are designed to show that the proposed clustering approach is capable of inferring global labels and reach performance comparable to the oracle. The results suggest that the gap between the meta-learning settings could be bridged.
>
> As discussed in the paper, we reiterate that *methods that have access to global labels are not directly comparable to those without global labels, since global labels provide much more information to the meta-learners*. This is evident from the observation that state-the-art methods almost all use feature pre-training. We will include more recent works (e.g. [2]) under the category of global-label methods.
>
> (continuing with the next comment)

---

> > ### Author Response · Authors · 2021-08-10
> > **Review Response (Part 2)**
> >
> > 3.2. **Narrowness of experiments**: We clarify that additional experiments on CIFAR-based datasets are included in the appendix. We agree that Meta-Dataset is a good candidate for exploring ill-defined global labels. However, the issue is beyond the current scope of the paper. (See our reply 2.2. above for more details.) On a practical note, meta-dataset is a huge dataset that is beyond the computational resources of many researchers (including this project).
> >
> > We clarify that the primary goal of our paper is to provide a theoretical understanding for the role of global labels in meta-learning. The other primary goal is to show that the two meta-learning settings could be bridged, which is supported by our experiments. We also explored various ablation scenarios to ensure that the clustering approach is viable for different datasets and sampling settings.
> >
> > We ran further experiments using a different architectures and obtained equivalent results. Please see the results below.
> >
> > ------------------------------
> > miniImageNet
> >
> > Resnet-12 99.9 (clustering accuracy)
> >
> > Resnet-18 99.9 (clustering accuracy)
> >
> > WRN-28 99.9 (clustering accuracy)
> >
> > tieredImageNet
> >
> > Resnet-12 96.4 (clustering accuracy)
> >
> > Resnet-18 95.8 (clustering accuracy)
> >
> > WRN-28  96.2 (clustering accuracy)
> >
> > ---------------------------------------------------
> >
> > The results show that using different architectures lead to comparable clustering outcomes and in turn similar test accuracies.
> >
> > We note that the labeler only uses the meta-representation features for clustering and any sufficiently expressive features learned in Phase 1 is usable. In addition, it would be computationally inefficient to train a larger model than needed for Phase 1. Further, Sec 5.2 and 5.3 already show that some miss-clustering (e.g 5% error) has very small impact on the overall model performance. Lastly, the final model (Phase 3) is standard supervised learning and could use any state-of-the-art architectures (e.g. our results in reply 3.1 (c) above  use WRN-28 network, as recommended by S2M2).
> >
> > The new hyper-parameters introduced in our work are used to determine the number of global clusters. This again reflects how much additional information is provided by global labels, since the hyper-parameters become unnecessary if we already know the number of global labels for a dataset.
> >
> > 3.3. **Computational Overhead**: Compared to methods leveraging feature pre-training, our method has a comparable two-stage process, since other methods perform pre-training + their main methods.
> >
> > For methods not using pre-training, our method trade-off computational complexity with noticeably higher performance (Table 1). We also note that our Phase 1 is designed to be light-weight, with a meta-representational formulation using fully differentiable local classifiers (ridge regression in Eq. 3).
> >
> > We thank the reviewer for his/her feedback and we’d be happy to answer any additional questions or clarify any of our answers further.

---

> > ### Comment · Reviewer_Cshp · 2021-08-29
> > **Follow-up**
> >
> > First and foremost, I would like to apologize to authors for engaging in the discussion that late in the decision-making process, and would like to thank them for the detailed answer/additional work. Below I try to follow up on some important points.
> >
> > 1.1  **Connection between global classification and meta-learning** : My point is that meta-learning is not the end-goal of FSL. The end-goal is to obtain a model that can produce discriminative features in a low-data regime. Meta-learning is just one way to tackle this problem, and as shown in the paper, there may exist others depending on the level of information your training data possesses: any option from fully unsupervised methods to global supervision is valid as long at it helps reach this goal. In that sense, I agree with the authors that *minimizing the meta-learning loss is precisely the goal of meta-learning, hence a naturally desirable outcome*, but that doesn't imply meta-learning is the best way to tackle few-shot learning.
> >
> > 1.2 **Connection between global classification and meta-learning**: This was not held against authors in my final rating, and is more of a suggestion for future revision.
> >
> > 2.1 **Class imbalance**: I thank the authors for the clarification and for the results.
> >
> > 2.2/3.2 **Ill-defined global labels and meta-dataset**: I do agree that this is an issue for methods that use global labels. In such regard, if given a dataset with no global labels at all, it is crystal clear than those methods won't be able to deal with this, while other *local* methods will be. Therefore, the working assumptions of each line of work is clear. The case of your method is slightly different, as your method would technically work when global labels are not present, but the question is: how would it compare to methods that do not explicitly try to find any such global labels ? It is not clear to me how whether the method would outperform *local* methods or degrade (because of the ill definition of global labels). I think this would have been a very relevant question to address, given that the motivation of the work relies on such situations existing in real life.
> >
> > Now on a practical note, reproducing the scenario of dataset merging is neither hard or computationally demanding. Meta-dataset contains 10 diverse datasets, 8 of which weigh < 15 GB. Designing a protocol to reproduce this scenario could help answer that question more precisely, and I believe would really strengthen the paper.
> >
> > 3.1 (a) **Comparison with self-supervised methods**: I do not believe a modified CMC would be similar to Proto-Net as it would not even be a meta-learner, but more a sort of standard metric-learning method where all positive samples (not notion of support or query) in the batch are pulled together and pushed from the rest. From a high level, I agree that Proto-Net also follows this idea, and so do other methods, but from a practical perspective, training and losses would be different. From the paper the results were taken from, the results of self-supervised are very closed to when trained in a fully supervised way, hence I do not expect that by adding more information (i.e local labels vs no labels at all), the results of this modified CMC would lag more than 10% behind the oracle, as does Proto-Net in your table.
> >
> > 3.1. (b) and 3.2 (c) **Data augmentation**: I agree with the point made by authors, as long as all methods reported are using exactly the same setting, the comparison is valid.

---

> > > ### Author Response · Authors · 2021-08-31
> > > **Response to Follow-up Questions**
> > >
> > > We thank the reviewer for engaging in the discussion. We address the concern below.
> > >
> > > **Regarding a comparison with methods that do not explicitly try to find global labels + experiments on Meta-dataset.**
> > >
> > > We designed a new protocol for testing on meta-dataset. Following the reviewer’s advice and *due to time constraints*, we tested on a subset of meta-dataset (including CUB, vgg_flower, and Aircraft datasets). Following the meta-dataset’s design, tasks are randomly sampled from each constituent dataset and only local labels are provided for each task. Meta-testing follows a similar procedure and evaluates on an identical number of tasks per constituent dataset.
> > >
> > > We compare our method against standard meta-representational learning (as described in our paper) and FRN [1], *a SOTA method that does not require pre-training with global labels*. We report the results averaged across three constituent datasets below:
> > >
> > > - Meta-representation Learning: 61.9 $\pm$ 0.5 (1-shot),  78.6 $\pm$ 0.4 (5-shot),
> > >
> > > - FRN: 63.1 $\pm$ 0.7 (1-shot), 79.7 $\pm$ 0.5 (5-shot),
> > >
> > > - Ours: 66.3 $\pm$ 0.5 (1-shot), 82.4 $\pm$ 0.3 (5-shot)
> > >
> > > The results show that also in this setting our method still outperforms local methods and doesn’t degrade the performance.
> > >
> > > For individual datasets, we observed that our method outperforms FRN noticeably on aircraft and vgg datasets, while it performed comparably to FRN on the cub dataset. Interestingly, we also note that the clusters inferred by our approach mix samples from different constituent datasets. *In other words, these clusters do not have any known global labels.*
> > >
> > > More broadly, we note that the notion of ill-defined global labels is a broad topic that covers many other possible scenarios and warrants investigation as a dedicated problem and *this is why we have identified it as relevant future works in our conclusion.* In our original experiments we addressed the scenario where the underlying global labels exist but are not accessible. Meta-dataset presents a further challenge compared to standard benchmarks when multiple datasets are used together.
> > >
> > > We would like to point out that the aim of our paper is *precisely* to highlight the issues related to global labels (e.g. the fact that they are ill-defined as being an issue that affects most state-of-the-art methods) and thus foster future research towards more challenging scenarios for meta-learning/few-shot learning where global labels are less clearly defined.
> > >
> > > **Regarding the connection between global classification and meta-learning.**
> > >
> > > While we agree that meta-learning is not the only approach to tackle FSL, we highlight that in the current literature, the *state-of-the-arts methods for FSL are predominantly meta-learning approaches*. In particular, we note that all SOTA papers cited by the reviewers are 1) meta-learning strategies that 2) use global labels for pre-training, which are *precisely the setting addressed in our paper*.
> > >
> > > Given such prevalence of using global labels in practice (in particular in SOTA meta-learning approaches), it is therefore very important to understand their contribution to overall performance in the context of meta-learning.
> > >
> > >
> > > **Comparison with CMC**: We thank the reviewer for the followup on CMC. We understand the reviewer’s point and agree that CMC is different from ProtoNet. Following the reviewer’s comment, we have studied the official implementation of CMC from [2], where we referenced the results from. The implementation (https://github.com/WangYueFt/rfs/blob/master/dataset/mini_imagenet.py, line 52 to 74) utilizes label information to construct training batches consisting of contrasting samples, which we think matches the reviewer’s idea of leveraging local labels. Therefore the results accurately reflect how CMC would perform in FSL. We will update our discussion on CMC in the empirical results.
> > >
> > > We thank the reviewer again for the followup questions. We hope that we our replies have addressed all the reviewer’s questions.
> > >
> > > [1] Wertheimer et al. CVPR 2021, Few-Shot Classification with Feature Map Reconstruction Networks
> > >
> > > [2] Tian et al. ICCV 2020, Rethinking Few-Shot Image Classification

---

> > > > ### Comment · Reviewer_Cshp · 2021-09-02
> > > > **Closing follow-up**
> > > >
> > > > I thank the authors for the efforts in producing those results. Here is my final follow-up on each point:
> > > >
> > > > **Experiments on meta-dataset**: I find the new results interesting, and I believe this scenario should be presented in the main paper (without attempting to tackle all possible scenarios of ill-defined labels, I think covering at least one of them as a proof that the method can actually still work is relevant). It is also quite interesting to see that cross-dataset clusters emerge, given that the chosen datasets have zero class or semantic overlap. A future iteration could explore more ill-defined combinations (for instance including MS-COCO and Aircraft as MS-COCO should contain the "airplane" class).
> > > >
> > > > **Comparison with CMC**: I thank the authors for this information. I am still however quite puzzled by the fact that their fully supervised ResNet-50 gets 57.6 % in 1-shot mini, while their simple supervised baseline with ResNet-12 gets 62.0 %. The CMC baseline being only 1% below the fully supervised ResNet-50 (and actually above in 5-shot) makes it reasonable to think that a properly tuned unsupervised baseline should be competitive.
> > > >
> > > > I have decided to increase my score to 6 as I believe authors have made a good job at addressing important concerns and providing new evidence during the rebuttal period. However, I will not go beyond as I am still not 100% convinced that proper comparisons were made (e.g CMC or other more recent metric learning / contrastive methods starting from the same meta-learnt pre-trained model for instance), and still not crystal clear about the overall message of the paper.

---

> > > > > ### Author Response · Authors · 2021-09-03
> > > > > **Thank you!**
> > > > >
> > > > > We thank the reviewer for the detailed feedback and for raising the score. It's much appreciated. We will incorporate your suggestions in our paper.

---

> ### Author Response · Authors · 2021-08-27
> **Follow-up**
>
> We thank the reviewer again for the feedback. With the discussion closing, if the reviewer has further questions, please let us know.

---

### Official Review · Reviewer_yjsY · 2021-07-16

**Rating:** 6
**Confidence:** 4

**Summary:**

This paper aims to reveal the effectiveness of using global labels in FSL from a theoretical perspective, and proposes a meta label learning (MeLa) method to infer the global labels by using a clustering algorithm. Overall, the investigated problem, i.e., whether to use global labels, is interesting and important. Also, the proposed theoretical analysis will be of interest to the community. However, the proposed clustering algorithm and the experimental part are not very surprising.

**Limitations And Societal Impact:**

Yes.

**Main Review:**

*Strengths
1. The investigated problem is interesting and important, which is of interest to the community.
2. This paper is well written and well organized.
3. The proposed theoretical analysis is important and inspiring.

*Weaknesses
I mainly have the following concerns about this paper:
1. It seems that some latest works are not reviewed and compared in this paper, e.g., FEAT [1] and DC [2]. In addition, some recent FSL works have paid attention to local representations, e.g., DN4 [3] and DeepEMD [4]. Are the theoretical analysis and conclusions still valid for these methods?
2. Why the data augmentation is avoided in the experiments? Since the experiments are mainly conducted on the image datasets, the reason of discarding data augmentation is not very clear.
3. How to construct the meta-training set?  It seems that the meta-training set is fixed in the paper. Do you randomly sample a series of meta-training tasks from the auxiliary set of the miniImageNet in advance and then fix it?
4. It will be interesting to explain whether the proposed theoretical analysis still works on the domain-shifted settings. As we know, the meta-training set has a somewhat different label space (even a different domain) with the meta-test set.

[1] Few-Shot Learning via Embedding Adaptation with Set-to-Set Functions. CVPR 2020.

[2] Free lunch for Few-shot Learning: Distribution Calibration. ICLR 2021.

[3] Revisiting Local Descriptor based Image-to-Class Measure for Few-shot Learning. CVPR 2019.

[4] DeepEMD: Differentiable Earth Mover's Distance for Few-Shot Learning. CVPR 2020.


================

Thanks for the authors' response. I would like to maintain my original rating of 6, and recommend the authors to further enhance the experiments according to the comments of all reviewers.



**Time Spent Reviewing:**

10 hours

---

> ### Author Response · Authors · 2021-08-10
> **Review Response**
>
> We thank the reviewer for the encouraging feedback. We have addressed all review concerns below.
>
> We emphasize that the proposed clustering method is designed to bridge the gap between two settings of meta-learning, the setting with access to global labels and the one without. Beyond the stated motivation that global labels may be unavailable, we further argue that using global labels is contrary to the goal of meta-learning: **global labels directly reveals how task samples relate to one another across tasks, while the aim of meta-learning is precisely to learn such relationships**. We will update our motivation to reflect this point.
>
> Therefore, while the proposed clustering method is simple, it is a crucial and necessary step to bridge the two meta-learning settings. On the other hand, the clustering approach in general led us to consider the novel and challenging setting of sample-without-replacement, where no same image appears across tasks (Section 5.3). It is surprising to us that meta-representation learned under this setting could still reliably infer the global labels. Additionally, we feel that it is not obvious that meta-learned  features ((Phase 1)) can be clustered reliably to infer global labels since Phase 1 features perform significantly worse than global classification features (Phase 3).
>
> **Detailed Clarification**:
>
> 1\. **Connection to [1] through [4]**: The theoretical analysis in our work demonstrates that features learned from global classification offer an expressive meta-representation suitable for few-shot learning. These features coincide precisely with those extracted in the pre-training step adopted in [1], [2] and [4], before applying the main methods from these works. The pre-training step contributes significantly to the overall model performance and it is difficult to disentangle the contribution from the pre-training vs the main methods. As an example, [2] is an extreme case whereby the pre-training contributes the most to the overall performance:
>
> [2] uses the pre-trained features obtained from S2M2, a global classification strategy (Mangla et al. 2020, “Manifold mixup for few-shot learning”). Below, we compare using the pre-trained features directly for few-shot learning (Eq. 7 in our paper) vs [2]:
>
> --------------------
> miniImageNet
>
> Pre-trained features: 66.5 $\pm$ 0.5 (1-shot), 83.8 (5-shot)
>
> [2]: 68.57 $\pm$ 0.55 (1-shot) 82.88 $\pm$ 0.42 (5-shot)
>
> CUB dataset
>
> Pre-trained features: 79.8 $\pm$ 0.5 (1-shot), 91.1 (5-shot)
>
> [2]: 79.56 $\pm$ 0.87 (1-shot), 90.67 $\pm$ 0.35 (5-shot)
>
> ------------------
>
> We note how, except for miniImageNet 1-shot, the pre-trained features are responsible for *all* the overall performance (with performance actually dropping after using the contribution from [2]). This comparison illustrates the difficulty of assessing many state-of-the-art methods that combine global classification and additional adaptation.
>
> Finally, we note that [3] does not use global labels or pre-training. As a result, the method obtains lower overall performance, which is in line with the findings in our paper highlighting the importance of global labels (see the discussion on conditional vs unconditional meta-learning methods in Section 3).
>
> 2\. **About data augmentation**: We focused on image classification tasks for our experiments since they are the most widely used benchmarks for meta-learning/few-shot learning. However, few-shot learning is applicable to various domains, including non-image input data (e.g. forecasting data), where it is not always possible to perform data augmentation (consider also the case of having the pre-trained feature representation of images as input, rather than the original images. These vector representations are difficult to augment).
>
> In addition, avoiding data augmentation also allows us to disentangle the performance contribution from each method vs auxiliary tricks that tend to improve performance significantly in computer vision. For the early meta-learning methods, data augmentation was not standard practice and Chen et al, “A Closer Look at Few-shot Classification” empirically validated that meta-learning methods performance was *significantly* underestimated for not using data augmentation.
>
> However, for completeness, we performed additional experiments using data augmentation, as requested by the reviewer. The results below demonstrate that global classification benefits significantly from data augmentation and outperforms/is competitive with most recent works. We followed S2M2, a global classification strategy (Mangla et al. 2020, “Manifold mixup for few-shot learning”) that uses extensive data augmentation.
>
> Comparison with recent state-of-the-arts
> -------------------------------------
> miniImageNet
>
> Ours: 66.5 $\pm$ 0.5 (1-shot), 83.8 $\pm$  0.2(5-shot)
>
> [1]:  66.78 (1-shot),  82.05 (5-shot)
>
> [2]: 68.57 $\pm$ 0.55 (1-shot) 82.88 $\pm$ 0.42 (5-shot)
>
> [3]: 51.24$\pm$0.74 (1-shot) 71.02$\pm$0.64 (5-shot)
>
> [4]: 66.50 $\pm$ 0.80 (1-shot) 82.41 $\pm$ 0.56 (5-shot)
>
> TieredImageNet
>
> Ours: 75.2 $\pm$ 0.4 (1-shot) 89.5 $\pm$ 0.3 (5-shot)
>
> [1]: 70.80 $\pm$ 0.23 (1-shot) 84.79 $\pm$ 0.16 (5-shot)
>
> [2]: 78.19 $\pm$ 0.25 (1-shot ) 89.90 $\pm$ 0.41 (5-shot)
>
> [3]: Nil
>
> [4]: 72.65 $\pm$ 0.31(1-shot) 86.03 $\pm$ 0.58 (5-shot)
>
> -------------------------------
>
> 3\. For presentation clarity and consistency we used the notation for fixed datasets in our theoretical analysis and algorithm. However, the same analysis could be easily extended to the standard setting where we sample from the meta-distribution (consistent with previous works). Our method is able to handle both scenarios.
>
> 4\. To the best of our knowledge, theoretical understanding on domain-shifting is an open problem and worth exploring in the future.
>
> We thank the reviewer for his/her feedback and we’d be happy to answer any additional questions or clarify any of our answers further.

---

> > ### Comment · Reviewer_yjsY · 2021-08-27
> > **Thanks for the response**
> >
> > Thank you for the responses and clarification!
> >
> > Most of my concerns have been addressed, and I will finalize my decision after discussing it with other reviewers.

---

### Official Review · Reviewer_CWrb · 2021-07-18

**Rating:** 5
**Confidence:** 4

**Summary:**

This paper tackles few-shot image classification problem which has two main categories of solution, meta-learning based methods and global classification based methods. The authors provide analysis to show that the few-shot result for global classification based method is the result upper bound for meta-learning based method. Due to the unavailability of global labels in global classification in real applications, the paper proposes one algorithm to convert local labels to global labels for global classification. Results show the effectiveness of the proposed method on miniIMAGENET and tieredIMAGENET datasets.

**Limitations And Societal Impact:**

The proposed method for converting local labels to global labels is only limited to be used in few-shot learning task.

The few-shot training pipeline is not carefully examined, with disjoint two stages in terms of classification weights.


**Main Review:**

This paper focuses on converting local labels to global labels, and then relies on global classification. The proposed Mela algorithm isolates the classification weights in meta-training and global classification, while reusing the weights as initialization weights for the other stage could potentially further improve the few-shot classification results. The training pipeline in Mela algorithm needs further examination.

The Algorithm 2 LearnLabeler, is essentially a clustering based approach with centroid assignment, update and pruning stages. Each stage has its defined metrics. Though this paper provides comparison with Self-Supervised methods (MoCo etc.) and  K-mean Clustering, as well as ablation studies for the effects of pruning threshold. However, the effects of some design details are still unclear. E.g. what if without the pruning stage? The effect of the metric selection (Equation 11 / Equation 12) in centroid assignment and update stages. How did you set the hyper-parameters in "K-mean Clustering" method?







**Time Spent Reviewing:**

4

---

> ### Author Response · Authors · 2021-08-10
> **Review Response**
>
> We thank the reviewer for the comments.
>
> We clarify that the primary goal of the paper is not “converting local labels to global labels”. Instead, our work explains theoretically and empirically validates the benefits of using global labels in meta-learning, which is a prevalent technique in practice.
>
> The other primary goal is to bridge the gap between two settings of meta-learning, the setting with access to global labels and the one without. The fact that the proposed clustering method achieves performance comparable to the oracle (i.e. classification with ground truth global labels) demonstrates its efficacy.
>
> While other clustering methods (and metrics) may be used, the specific choice of clustering algorithm does not affect the intended observation that clustering plus local constraints are capable of inferring global labels reliably.
>
> **Detailed Clarifications**:
> 1. We do not re-use weights from meta-learning (Phase 1) as initialization weights for other models. The model from Phase 1 is only used for clustering samples and deriving a labeler. Phase 3 training starts from random initialization.
>
> 2. **No Pruning**: The pruning threshold and number of initial clusters controls the number of final clusters. Table 2 already shows that the proposed method is robust to a reasonable range of final cluster counts. Too few or too many final clusters would negatively affect the final performance.
>
> 3. **Metric selection**: L2 distance is a standard metric for clustering and it performs very well in our experiments. As discussed above, investigating other clustering methods would not affect our thesis.
>
> 4. **Hyper-parameter for K-Mean**: The comparison between the proposed clustering approach vs K-mean is to illustrate the importance of leveraging local information for inferring global labels. As stated in the paper, the number of clusters in k-Mean is set to the actual number of global classes present in the dataset. The setup in fact favors K-mean (the competitor method) since it is the optimal number of clusters.
>
> We thank the reviewer for his/her feedback and we’d be happy to answer any additional questions or clarify our answers further.

---

> > ### Comment · Reviewer_CWrb · 2021-08-30
> > **Reviewer response after rebuttal**
> >
> > My original reviews also have concerns in the aspects of model novelty (clustering + pruning) and performance comparison with SOTA. However, the authors argues that “The other primary goal is to bridge the gap between two settings of meta-learning, the setting with access to global labels and the one without.” From this aspect, the whole paper did achieve this claim.
> >
> > Per the authors' response: "No Pruning: The pruning threshold and number of initial clusters controls the number of final clusters. Table 2 already shows that the proposed method is robust to a reasonable range of final cluster counts. Too few or too many final clusters would negatively affect the final performance." Table 2 does show the results for a reasonable range of final cluster counts (q: 4-6.5), and the results are stable. Do the authors have specific results to show that "Too few or too many final clusters would negatively affect the final performance"? What if we don't have the pruning part?

---

> > > ### Author Response · Authors · 2021-08-31
> > > **Clarification For Reviewer Response**
> > >
> > > We thank the reviewer for providing additional feedback. We address the reviewer’s concerns below.
> > >
> > >
> > > **Regarding the paper’s objectives.**: we appreciate that the reviewer agrees that we achieved our goal of bridging the gap between the two settings. We further clarify on why this goal is important: using global labels (e.g. feature pre-training) has become a standard practice in meta-learning/few-shot learning. Extensive empirical evidences (e.g. the SOTA papers cited by other reviewers) show that leveraging feature pre-training with global labels significantly improve model performance. Therefore it is important 1) to understand theoretically the role of global labels (our other primary contribution, Sec. 3 of the paper), and 2) bridge the gap between the two settings (with or without global labels). We will further clarify this in the paper.
> > >
> > > **Regarding further comparison with SOTA.**: We were not aware this was a concern of the reviewer from the initial feedback. Please note that we added a comparison with recent SOTA methods as requested by other reviewers. We include the results below:
> > >
> > > *miniImageNet*
> > >
> > > **Ours**: 66.5 $\pm$ 0.5 (1-shot), 83.8 $\pm$  0.2(5-shot)
> > >
> > > [1]: 66.50 $\pm$ 0.80 (1-shot) 82.41 $\pm$ 0.56 (5-shot)
> > >
> > > [2]:  66.78 (1-shot),  82.05 (5-shot)
> > >
> > > [3]: 65.9 (1-shot) and 82.4 (5-shot)
> > >
> > > *TieredImageNet*
> > >
> > > **Ours**: 75.2 $\pm$ 0.4 (1-shot) 89.5 $\pm$ 0.3 (5-shot)
> > >
> > > [1]: 72.65 $\pm$ 0.31(1-shot) 86.03 $\pm$ 0.58 (5-shot)
> > >
> > > [2]: 70.80 $\pm$ 0.23 (1-shot) 84.79 $\pm$ 0.16 (5-shot)
> > >
> > > [3]: 71.2 (1-shot )and 86.0 (5-shot)
> > >
> > > The results suggest that our method outperforms/is competitive with SOTA methods. The new results exploit practical techniques (e.g. data augmentation) to further improve model performance.
> > >
> > > **Regarding the question on using too many or too few clusters or no pruning at all.** We include the results below for *miniImageNet*. Due to time constraints, we will have to defer results for tieredImageNet to the final version of the paper.
> > >
> > > - 48 Clusters: 58.7 ± 0.4 (1-shot), 75.8 ± 0.3 (5-shot)
> > > - 250 Clusters: 59.2 ± 0.4 (1-shot), 75.2 (5-shot)
> > >
> > > The results show that significantly over/under-estimating the number of clusters does negatively affect overall performance.
> > >
> > > No pruning is feasible if we slightly modify our clustering strategy. Instead of using a simple random initialization (our current routine), we can use a more advanced one (k-means++ initialization). In this case, we only have to set one hyper-parameter, namely the desired number of clusters (since no pruning is used). We report the results below.
> > >
> > > - 64 Clusters: 60.0 ± 0.4 (1-shot), 77.6 ± 0.3 (5-shot)
> > > - 58 Clusters: 59.5 ± 0.5 (1-shot), 76.9 ± 0.4 (5-shot)
> > >
> > > The results show that the no-pruning scenario is comparable to our current strategy, which achieved 60.2 ± 0.4 (1-shot) and  77.6 ± 0.3 (5-shot). We note that for miniImageNet, real labels could be recovered near perfectly.
> > >
> > > Lastly and importantly, we note that the hyper-parameters for both our current clustering method or the no-pruning scenario are **determined by the meta-validation set**. This prevents our algorithm from significantly over/under-estimate the number of actual global clusters.
> > >
> > > We hope that the additional quantitative results addressed the reviewer’s all remaining questions and that the reviewer might consider revising our paper’s final score.
> > >
> > > [1] Zhang et al. CVPR 2020, DeepEMD: Few-Shot Image Classification with Differentiable Earth Mover’s Distance and Structured Classifiers
> > >
> > > [2] Ye et al. CVPR 2020, Few-Shot Learning via Embedding Adaptation with Set-to-Set Functions
> > >
> > > [3] Wertheimer et al. CVPR 2021, Few-Shot Classification with Feature Map Reconstruction Networks

---

> > > > ### Comment · Reviewer_CWrb · 2021-09-03
> > > > **Final**
> > > >
> > > > I increase the score a bit. But, remember that, the proposed model in Section "4 Meta Label Learning" consists of two parts: Clustering Step and Pruning Step. However, now the authors show that "The results show that the no-pruning scenario is comparable to our current strategy." This further simplifies the model, and weakens the novelty of the proposed model.

---

> > > > > ### Author Response · Authors · 2021-09-03
> > > > > **Thank you!**
> > > > >
> > > > > We thank the reviewer for the feedback and raising the score. It's much appreciated.

---

### Public Comment · ~Brando_Miranda1 · 2022-04-21
**Difference between MeLA (using local labels to get inferred global labels) vs Unsupervised Learning vs Contrastive Learning**

First interesting work! thanks for sharing & congrats on acceptance.

I am curious, how is this work related to unsupervised learning, self-supervised learning & contrastive learning? I understand that if local labels are present e.g. as they often are in a episodic meta-learning way -- that MeLA can be applied.

However, I thought it would be very interesting for a comparison with previous related work -- especially unsupervised like vision learning algorithms.

In particular, the reason I believe this is interesting is, if we can generate local labels, then why not use that to produce global labels and skip MeLa? e.g. Clip and other internet vision-NLP data allows us to do things like that easily and get very good foundation like models for vision.

PS: never made a comment here like this before so hopefully I get some type of response on my email? in case my email is brandojazz AT gmail DoT c o m

---

### Public Comment · ~Brando_Miranda1 · 2022-04-21
**How is it that if the global labels upper bounds the meta-learning loss -- that we don't get pre-training to be empirically worse on the meta-test?**

Hi Authors!

I was thinking about the results you had and had the following questions:

1. Why do you observe that global labels beat meta-learning if according to the results they both optimize the same objective?
2. Related to 1, how is it that pre-training beats meta-learning given that you only show an upper bound? (even when the loss is zero in never goes bellow the meta-learning objective as far as I understand)

Did I misunderstand the paper's main contributions, perhaps? The contribution says:

> In this paper, we show why exploiting pre-training is theoretically advantageous for meta-learning, and in particular the critical role of global labels.

but what is shown is the relation between global labels and the meta-learning objective, not that global labels is critical (not sure what that means formally).

Thanks for your time and regardless of what my questions might imply, I found it very really fascinating work!

PS: never made a comment here like this before so hopefully I get some type of response on my email? in case my email is brandojazz AT gmail DoT c o m

---

> ### Public Comment · Authors · 2022-04-22
> **Thank you for the interest**
>
> Hi Brando,
>
> Thank you for the interest in our work. Please see the clarification below.
>
> The key insight here is that using global labels induces a *conditional* meta-learning problem, where the global labels are useful info to improve learning, compared to more classical meta-learning approaches that are unconditional (e.g. MAML). Specifically, use of global labels is equivalent to having a inner algorithm W[Y] where the global labels Y directly identifies the task classifier, rather than learned via optimization.
>
> Theoretical results showing why conditional meta-learning is better than unconditional ones are from a previous paper: "The Advantage of Conditional Meta-Learning for Biased Regularization and Fine-Tuning". This was discussed in our paper too.
>
> The proof also shows that the ability to classify more classes is a more difficult problem (where the upper bound comes from).  Reducing the number of classes being classified during the meta-testing phase (e.g. 64-way during pre-training to 5-way during testing) would improve the classification loss. This is also reflected in improved empirical accuracy.
>
> More practically, W[Y] can be interpreted as a regularization where classification vector W[y] for each class y is shared across all tasks.

---

> > ### Public Comment · ~Brando_Miranda1 · 2022-04-22
> > **Friendly follow up: Don't understand (or agree) that MAML is unconditional**
> >
> > Hi Authors!
> >
> > Thank you for the prompt response and congrats again on the very interesting work! Wish we had more work going in this direction.
> >
> > What I'm trying to understand is the scope of the conclusions that your work captures (including previous work of yours, which I'm familiar and find interesting). In particular, I currently believe it does not apply to MAML -- or any meta-learning algorithm that has a universal approximator (neural net) -- because as far as I understand, there is no argument to involve neural networks in the entire process or analysis (starting bias induced by the approximation error wrt hypothesis class of compositional functions, to the optimization error to the final generalization error). In contrast to work that has bounds but does include the neural network: Generalization Bounds For Meta-Learning: An Information-Theoretic Analysis https://proceedings.neurips.cc/paper/2021/hash/d9d347f57ae11f34235b4555710547d8-Abstract.html
> > Also, in contrast to work that does the entire analysis is work by theoretician Matus Telegarsky, Tomaso Poggio or Maxim Raginsky. How does your work compare with theirs? How does your work theoretically include the fact that the meta-learning might be done by the Neural Network? Also, if I understand correctly, the analysis/conclusions depends on the fact that we have a fixed embedding -- it does not analyze how the embedding method was obtained or what happens if we use **different** embedding methods (e.g. different architectures, meta-learners, way to obtain the weights given a fix arch etc).
> >
> > Also, consider the following to show that MAML might be conditional and unconditional at the same time -- depending on how you write it (according to your notation).
> >
> > So we have (with your notation):
> >
> > $$ \theta = \min_{\theta \in R^n} E_{\mu, s \sim \rho} E_{S \sim \rho^n} L_{\mu}(A(\tau(s), S) ) $$
> >
> > In your case you have A to be restricted to Ridged regression (which has a closed form). If instead we have A to be MAML how would it look like?
> > MAML can be written in a Conditional way:
> > $$ A(\tau(s), S) = f_{\theta - \eta \nabla L_{S}}( \cdot) $$
> > Where $\tau(s) = \theta - \eta \nabla L_{s}$
> > Note, the side information is a data set. I know you do discuss this specific scenario and you don't allow that Z=s. So in that case just split the support set (or the query and support) such that we have data to adapt. You mention that is possible and that is what I am suggesting you do and it's crucial to analyze this scenario in depth with the entire process in the loop -- in particular the role of the neural network.
> > At the core my question would basically be how do you know that a few gradient steps do not condition enough -- especially considering the entire meta-training process -- so that it's a conditional meta-learner? In addition how do you know that even if this adaptation is too weak to truly condition, that however weights for the NN learned **combined with the fact you are passing this through a neural net and it was trained using that neural net from the start** that it won't just output the conditional optimal predictor? e.g. see the appendix of some work I've done very basic work attempting to do this: https://arxiv.org/pdf/2112.13121.pdf where if the task information is given then we can output the optimal predictor if the predictor is arbitrarily powerful. My analysis has many obvious flaws, like it assume the unknown distribution is known but a strength is that it does consider the entire process from start to finish.
> >
> > MAML can be written in a Unconditional way (but it's not actually unconditional):
> > Consider this:
> > MAML can be written in an Unconditional way:
> > $$ A(\theta, S) = f_{\theta - \eta \nabla L_{S}}( \cdot) $$
> > this is actually the same as the previous meta-learner that I wrote in the previous paragraph. Thus, the unconditional scenario would suggest that it can get the same error as the conditional because MAML has a neural network involved with an adaptation that takes in task information.
> >
> >
> > I know this was a really long response and I apologize -- but I think it's important to clarify how MAML fits into your framework and if the analysis you provide really applies to MAML (which I don't believe it does, unless you respond to some of my questions or at least relate your work with Qi Chen's et al's Generalization Bounds For Meta-Learning: An Information-Theoretic Analysis.
> >
> > Also, I think the analysis should have been with this objective:
> >
> > $$ f= \min_{f \in F} E_{\mu, s \sim \rho} E_{S \sim \rho^n} L_{\mu}(f(S,s)) $$
> >
> > Also, final comment, the meta-learner you propose is ridge regression (special case of Tikhonov for l2):
> >
> > $$ A(\tau(s), S) = arg\min_{w \in R^n} \frac{1}{n} \sum^n_{i=1} l(<w, x>) + \lambda \| w - \tau(s) \|^2_2$$
> >
> > which is the "closest" thing to a "linear meta-learner" in the meta-learner work. It does not mention how the features were obtained.
> >
> > Thank you again for your generous time in discussing this with me! I find this line of work important for us to make good thoughtful process in meta-learning.
> >
> > I've been doing a more in depth literature search and think this work is important to put into context with each other (google doc link https://docs.google.com/document/d/1ajkHQWBq5H6lhduU6903jGTG9-lQUQDx2fpmmyBTmoA/edit?usp=sharing):
> >
> > The Advantage of Conditional Meta-Learning for Biased Regularization and Fine-Tuning:
> > NeurIPS version: https://proceedings.neurips.cc/paper/2020/file/0a716fe8c7745e51a3185fc8be6ca23a-Paper.pdf
> > Video: https://crossminds.ai/video/the-advantage-of-conditional-meta-learning-for-biased-regularization-and-fine-tuning-606feec7f43a7f2f827c1515/
> > The Role of Global Labels in Few-Shot Classification and How to Infer Them
> > Paper: https://arxiv.org/abs/2108.04055
> > Open review: https://openreview.net/forum?id=3S0z0IjWkyl
> > NeurIPS video: https://slideslive.com/38968089/the-role-of-global-labels-in-fewshot-classification-and-how-to-infer-them?ref=speaker-17625-latest
> > Generalization Bounds For Meta-Learning: An Information-Theoretic Analysis
> > Video: https://slideslive.com/38968315/generalization-bounds-for-metalearning-an-informationtheoretic-analysis?ref=recommended
> > Paper: https://arxiv.org/pdf/2109.14595.pdf
> > Information-Theoretic Generalization Bounds for Meta-Learning and Applications
> > https://www.mdpi.com/1099-4300/23/1/126/pdf
> > Modeling and Optimization Trade-off in Meta-learning
> > https://arxiv.org/pdf/2010.12916.pdf
> > Large-Scale Meta-Learning with Continual Trajectory Shifting
> > https://arxiv.org/pdf/2102.07215.pdf
> > Unraveling Meta-Learning: Understanding Feature Representations for Few-Shot Tasks
> > https://arxiv.org/pdf/2002.06753.pdf

---

> > > ### Public Comment · Authors · 2022-04-23
> > > **Follow-up Clarification**
> > >
> > > Thanks for the thoughtful discussion. Please see the clarifications below.
> > >
> > > **Why the neural network learning process is not included in our analysis**: There has been existing works showing that effective feature reuse, rather than the adaptation process, contributes most of the performance. On MAML specifically, please see the discussion from [1].
> > >
> > > Other than MAML, there are also many meta-learning methods that work with a fixed embedding during the adaptation process (e.g.[1], R2D2 [2], ProtoNet etc). In this class of methods, the key question is how to obtain the robust embedding for adaptation during meta-testing, since the embedding would remain fixed during the testing. Thus we focused on the fixed embedding scenario in the paper.
> > >
> > > **Conditional meta-learning**: It is true that MAML could be written in a conditional way, but in your formulation, the conditioning is the same as the inner algorithm, which would be hard to interpret as conditioning. Empirically, [1] also demonstrated that even during the *full network adaptation*, the embedding features remain stable and only the linear classifier on top is learned.
> > >
> > > In addition, we have discussed in the paper that in the pre-training case, global labels are a much more powerful side-info to use, compared to the dataset. The global labels directly reveals how all tasks relate to one another unambiguously. One of the goals for meta-learning is precisely trying to understand how past tasks relate to each other so that shareable knowledge could be extracted and used for future tasks. Thus we believe that global labels provide stronger side-info. As mentioned in our previous response, the global labels forces related tasks to share task classifiers (each class y uses a shared classification vector W[y]), which may be interpreted as how global labels regularise the training.
> > >
> > > **On obtaining the embedding**: In line with other meta-learning methods that use fixed embedding during meta-testing, the training process is optimising the embedding with the inner algorithm considered in the loop. In our case, since the conditional inner algorithm is simply W[Y], it just corresponds to standard multi-class classification.
> > >
> > > [1]. Rapid Learning or Feature Reuse? Towards Understanding the Effectiveness of MAML, Raghu et al. ICLR 2020
> > >
> > > [2]. Meta-learning with differentiable closed-form solvers, Bertinetto et al.,  ICLR 2019

---

### Decision · Program_Chairs · 2021-09-27

**Decision:**

Accept (Poster)

**Comment:**

The main issues amongst reviewers were 1. A lack of compelling scenarios without global labels, 2. Experiments are only on small benchmarks (in today’s environment) in which global labels are all well-defined. The main suggestion was to apply the technique to the meta-dataset benchmark, and the authors have provided some initial results that are indeed promising, however very preliminary. Other issues like performance with data augmentation, class imbalance (robustness in general) have been satisfactorily addressed in the discussion period.

The reviewers tend to agree that a more in-depth look at the case with no global labels is a topic for future research. Even still, there is quite a bit of work to be done to incorporate the new results/feedback from the discussion period, and the reviewers remain neutral-to-positive. I do think that the hypothesis and associated performance is compelling enough to be of interest to the community. As a last suggestion, I think it’s worth briefly citing and discussing the paper “Unsupervised Learning Via Meta-Learning” by Hsu et al., ICLR 2019. The approaches are somewhat different, but related enough to be worth mentioning.